


# Modelling past, present and future peatland carbon accumulation across the pan-Arctic

**Nitin Chaudhary, Paul A. Miller and Benjamin Smith**

Department of Physical Geography and Ecosystem Science, Lund University,
Sölvegatan 12, SE- 22362 Lund, Sweden

*Correspondence to:* **N. Chaudhary (nitin.chaudhay@nateko.lu.se)**

**Abstract.** Most northern peatlands developed during the Holocene, sequestering large amounts of carbon in terrestrial ecosystems. However, recent syntheses have highlighted the gaps in our understanding of peatland carbon accumulation. Assessments of the long-term carbon accumulation rate and possible warming driven changes in these accumulation rates can therefore benefit from process-based modelling studies. We employed an individual- and patch-based dynamic global ecosystem model with dynamic peatland and permafrost functionality and vegetation dynamics to quantify long-term carbon accumulation rates and to assess the effects of historical and projected climate change on peatland carbon balances across the pan-Arctic. Our results are broadly consistent with published regional and global carbon accumulation estimates. A majority of modelled peatland sites in Scandinavia, Europe, Russia and Central and eastern Canada change from carbon sinks through the Holocene to potential carbon sources in the coming century. In contrast, the carbon sink capacity of modelled sites in Siberia, Far East Russia, Alaska and western and northern Canada was predicted to increase in the coming century. The greatest changes were evident in eastern Siberia, northwest Canada and in Alaska, where peat production, from being hampered by permafrost and low productivity due the cold climate in these regions in the past, was simulated to increase greatly due to warming, wetter climate and greater $CO_2$ levels by the year 2100. In contrast, our model predicts that sites that are expected to experience reduced precipitation rates and are currently permafrost free will lose more carbon in the future.

## 1 Introduction

The majority of the northern peatlands developed during the Holocene ca. 8-12 thousand years (kyr) ago, after the deglaciation of the circum-Arctic region (MacDonald et al., 2006). The availability of new land surfaces owing to ice retreat (Dyke et al., 2004; Gorham et al., 2007), climate warming following deglaciation (Kaufman et al., 2004), increased summer insolation (Berger and Loutr, 2003), more pronounced seasonality (Yu et al., 2009), greenhouse gas emissions (MacDonald et al., 2006) and elevated moisture conditions (Wolfe et al., 2000) are some of the factors that promoted the rapid





expansion of northern peatlands. Moderate plant productivity together with depressed decomposition due to saturated conditions led to a surplus of carbon (C) input relative to output, resulting in the accumulation of peat (Clymo, 1991). Peatlands of the Northern Hemisphere are estimated to have sequestered approximately 350-500 PgC during the Holocene (Gorham, 1991; Yu, 2012).

Peatlands share many characteristics with upland mineral soils and non-peat wetland ecosystems. However, they constitute a unique ecosystem type with many special characteristics, such as a shallow water table depth, organic, C-rich soils, a unique vegetation cover dominated by bryophytes (hereinafter referred to as "mosses"), spatial heterogeneity, anaerobic biogeochemistry and permafrost in many regions. Due to their high C density, and the sensitivity of their C exchange with the atmosphere to
temperature changes, these systems are an important component in the global C cycle and the coupled Earth system (MacDonald et al., 2006). Lately, considerable effort has been put to incorporating peatland accumulation processes into models, with the purpose of understanding the role of peatlands in sequestering C, thereby lowering the radiative forcing of past climate (Frolking and Roulet, 2007; Wania et al., 2009a; Frolking et al., 2010; Kleinen et al., 2012; Tang et al., 2015), and how they might
affect future climate warming and the C cycling (Ise et al., 2008; Swindles et al., 2015).

Clymo (1984) developed a simple one-dimensional peat accumulation model and described the main processes and mechanisms involved in peat growth and its development. This model became the starting point for later work in many peat growth modelling studies. Hilbert et al. (2000) developed a
theoretical peat growth model with an annual step, modelling the interaction between peat accumulation and water table depth using two coupled non-linear differential equations. Using a similar approach, Frolking et al. (2010) developed a complex Holocene peat model by combining the dynamic peat accumulation model of Hilbert et al. (2000) with a peat decomposition model (Frolking et al., 2001). They showed that the model performed fairly well in simulating the long-term peat accumulation,
vegetation and hydrological dynamics of a temperate ombrotrophic bog in Ontario, Canada. Though the models mentioned above are detailed enough to capture the peat accumulation and decomposition processes quite robustly, they lack soil freezing-thawing processes, and this limits their application over regions where such processes occur. Wania et al. (2009a) were first to account for peat dynamics in a model for large-area application, incorporating peatland functionality in the LPJ-DGVM model,
designed for regional and global simulation of ecosystem responses to climate change (Sitch et al., 2003). Their approach included a number of novel features - such as a detailed soil freezing-thawing scheme, peatland-specific plant functional types (PFTs) and a vegetation inundation stress scheme - but employed a two-layer representation of the peat profile, which is not as detailed as the process-based dynamic multi-layer approaches taken by Hilbert et al. (2000) and Frolking et al. (2010).



Though much information is available about the past and present rates of C accumulation in the literature, recent syntheses have highlighted the existing spatial gaps in data availability across the pan-Arctic (45-75 °N) (Yu et al., 2009; Loisel et al., 2014). The extent and remoteness of many locations present challenges for the reliable estimation of total C, basal ages and C accumulation rates (CAR). In the present study, we use the individual- and patch-based dynamic global ecosystem model LPJ-GUESS

(Smith et al., 2001), a climate-driven model of peat accumulation across the pan-Arctic under past, present and future climate. The model is extended to include dynamic peatland functionality and vegetation dynamics as described in Chaudhary et al. (2016) to quantify the spatial and temporal C accumulation rates (CAR) and to assess the potential effects of historical and projected climate and atmospheric $CO_2$ on peatland C balances at regional scale across the pan-Arctic.

## 2 Methodology

### 2.1 Model description

LPJ-GUESS (Lund-Potsdam-Jena General Ecosystem Simulator) is a process-based model of vegetation dynamics, plant physiology and the biogeochemistry of terrestrial ecosystems. It simulates vegetation structure, composition and dynamics in response to changing climate and soil conditions based on an individual- and patch-based representation of the vegetation and ecosystems of each simulated grid cell and is optimized for regional and global applications (Smith et al., 2001; Sitch et al.,

2003; Miller and Smith, 2012). The model has been evaluated in comparison to independent datasets and other models in numerous studies; see e.g. Piao et al. (2013), McGuire et al. (2012), Smith et al. (2014) and Ekici et al. (2015).

We employed a version of LPJ-GUESS customised for studies of Arctic ecosystems (Miller and Smith,

2012) that has been developed to include dynamic, multi-layer peat accumulation functionality and permafrost dynamics. The model represents the major physical and biogeochemical processes in upland and wetland arctic ecosystems, including an expanded set of plant functional types (PFTs) specific to these areas (McGuire et al., 2012; Miller and Smith, 2012). The revised model is described in outline below, while a full description can be found in Chaudhary et al. (2016). In our approach, vegetation and

peatland C dynamics are simulated on multiple, connected patches to account for the functional and spatial heterogeneity in peatlands. The simulated PFTs have varied structural and functional characteristics and can establish in each connected patch and compete for soil resources, area and light. The plant population and structure of the plant community is an emergent outcome of this competition. The model is initialised with a random surface constituting 10 patches of uneven height. Heterogeneity

in height of adjacent patches is a precondition for hydrological redistribution between them, which mediates vegetation succession and affects the peat accumulation rate, as described below. The soil/peat





column is represented by four different vertically-resolved layers. A dynamic single snow layer on overlays the peat column, represented by a dynamic litter/peat layer consisting of a number of sublayers, updated yearly, that depends on its thickness. Underneath the peat column is a fixed 2 m deep

mineral soil column consisting of 0.1 m thick sublayers, which is underlain by a 48 m deep "padding" column consisting of relatively thicker sublayers. The soil temperature is updated daily for each sublayer at different depths enabling the simulation of a dynamic soil thermal profile, a basis for the representation of permafrost (Wania et al., 2009a). The fractions of ice and water as well as the mineral and peat fractions in each layer govern the heat capacities and thermal conductivities and affect freezing

and thawing processes of soil water in peat and mineral soil layers (Wania et al., 2009a). The fractions of water and ice in the sublayers is updated each day depending upon the variation in the soil temperature and fractional mineral content, following Hillel (1998). A detailed description of permafrost and soil temperature scheme is available in Chaudhary et al. (2016) and Miller and Smith (2012) and references therein.

A water bucket scheme was used to simulate peatland hydrology where the assumption is made that the precipitation is the main input of water. Evapotranspiration, drainage, surface and base runoff are the major water balance processes in the peat layers (Gerten et al., 2004). The model also includes lateral flow of water between patches, an important governing process of vegetation and C dynamics of

peatlands that is lacking in most peatland models. From higher elevated patches (hummocks) to lower depressions (hollows), the water flows using a simple lateral flow scheme. In this scheme, water table position (WTP) of individual patches is reset to the mean landscape WTP in each daily time-step. Elevated patches lose water while the depressed patches gain water with respect to the mean landscape WTP, affecting plant productivity and decomposition rates in each patch and resulting in dynamic

surface conditions over time.

Five PFTs are used to represent the main functional elements of peatland vegetation: graminoids (Gr), mosses (M), deciduous high shrubs (HSS) and deciduous and evergreen low shrubs (LSS and LSE). PFTs differ in physiological, morphological and life history characteristics that govern their interactions

and responses to climate and evolving system state. Key PFT parameters in the present study include C allocation, phenology, rooting depth, tolerance of waterlogging, and decomposability of PFT-derived litter (Miller and Smith, 2012). Prescribed bioclimatic limits (Miller and Smith, 2012) and favoured annual-average WTP (aWTP) ranges determine the PFTs' establishment and mortality (see Table A1) and reflect their distribution ranges. Shrubs are favoured in dry conditions (Malmer et al., 2005) where

aWTP is below -25 cm (we use a sign convention in which a negative value of WTP signifies a water table below the peat surface). Conversely, mosses and graminoids are more vulnerable to dry conditions. Graminoids favour saturated conditions and establish when aWTP is above -10 cm while mosses establish when the aWTP is between +5 and -50 cm. The establishment function is implemented



annually and dependent on aWTP.

The balance between the annual addition of new litter layers on top of the mineral soil column and the daily decomposition rate governs the peat accumulation. C originating from different PFTs accrues as litter in the peat layers at variable rates depending on the differences in PFT mortality, productivity and leaf turnover. The accumulated peat decomposes on a daily time step based on the plant litter types in each layer of a patch with decomposition rates that are controlled by soil physical and hydrological properties in each layer. Differences in the peat decomposition rates among PFTs arise from their intrinsic properties and structure, parameterized using an initial decomposition rate $k_o$ (see Table A1) (Aerts et al., 1999; Frolking et al., 2001; Chaudhary et al., 2016) which is assumed to decline over time (Clymo et al., 1998).

The way plant access water from the mineral soil and dynamic peat layers in each patch necessitated a readjustment of the soil layer representation relative to the standard version of LPJ-GUESS, taking into account the depth of dynamic peat layers and the mineral soil layers. In the modified water uptake scheme, there are two static underlying mineral soil layers: an upper mineral soil (UMS) layer and a lower mineral soil (LMS) layer, of 0.5 and 1.5 m depth respectively. In the absence of peat, the fraction of roots in these two layers is prescribed for different PFTs (Smith et al., 2001) and determines the daily plant uptake of water from the mineral soil (Table A1) (Chaudhary et al., 2016). We prescribed rooting depth fractions of 0.7 and 0.3 to UMS and LMS, respectively, for shrubs while graminoids are assumed to have relatively shallow rooting depths, with fractions of 0.9 and 0.1 in the UMS and LMS layers respectively (Bernard and Fiala, 1986; Malmer et al., 2005; Wania et al., 2009b). During the initial stages of peat accumulation, plant roots are still present in both in UMS and LMS layers but their root proportion declines linearly as peat builds up while a greater root fraction is added to the peat layers to access water from the peat soil. Mosses are assumed to take up water from the top 50 cm of peat (Shaw et al., 2003; Wania et al., 2009b) once peat height exceeds 50cm. Before this mosses take water only from the mineral soil layers. All other PFTs can take up water from both mineral soil layers and peat layers until peat height reaches 2 m, after which they can only access water from the peat soil layers.

## 2.2 Simulation Protocol and Data Requirements

### 2.2.1 Hindcast experiments

To initialise the model with vegetation in equilibrium with early Holocene climate, the model was run from bare ground surface conditions for the first 500 years, recycling the first 30 years of the Holocene climate data set (see below), The mineral and peat layers were forced to remain saturated for the entire initialisation period. The peat decomposition, soil temperature and water balance calculations began





when the peat column reached a minimum thickness of 0.5 m. We adopted this model initialisation strategy to avoid sudden collapse of the peat column in very dry conditions because shallow peat can become drier or wetter within a very short span and continuous dry periods would increase temperature dependent decomposition rates and reduce the accumulation rate markedly.

To adequately represent the peatland history and dynamics across the major bioclimatic domains of the pan-Arctic, the model was applied at 180 grid points (referred to as 'sites' below) distributed among 10 geographical zones spanning the circum-Arctic from 45-75 °N (Fig. 1), each zone being represented by 10-20 randomly selected points, (see Fig. 1 and Table 1). Each simulation was run for 10,100 years, and
195 comprised three distinct climate forcing periods. The first, Holocene phase, lasted from 10 kyr before present (BP) until 0 BP, taken to correspond to the year 1900. Details of the source and preparation of the Holocene climate forcing data are explained in Chaudhary et al. (2016). The second, historical phase ran from 1901 until 2000. Finally, the future scenario phase (see Section 2.3.2) ran from 2001 until 2100. We forced the model with daily climate fields (temperature, precipitation and cloudiness)
constructed by interpolating between monthly values from 10 kyr BP until 1900, after which the CRU climate data set (Mitchell and Jones, 2005) was used for the historical phase of the simulation until the year 2000. Millennial atmospheric $CO_2$ concentration values from 10 kyr BP to 1850 AD used as a boundary condition in the Hadley Centre Unified Model (UM) (Miller et al., 2008) were linearly interpolated to yearly values and used to force LPJ-GUESS. Observed annual $CO_2$ values from
atmospheric or ice core measurements were used, to force LPJ-GUESS from 1850 to 2000 (McGuire et al., 2012).

Accurate prediction of total C accumulation at any particular location depends on selecting the right inception period, the C content and lability of the peat material, its bulk density over time and depth, as
well as local hydro-climatic conditions (Clymo, 1992; Clymo et al., 1998). Bulk density and C fraction values vary widely among different peatlands, and reliable estimates are often lacking (Clymo et al., 1998). Basal ages are the proxies of peatland initiation history, are often hard to determine and are not available for many key peatland types. For example, Eastern Siberia and European Russia are regions that have not been well-studied in this regard (Loisel et al., 2014; Yu et al., 2014a). We therefore started
simulations at the same time (10 kyr BP) for all 180 sites and fixed initial bulk densities to 40 kg C m$^{-3}$.

We calculated rate of C accumulation as long-term (apparent) rate of C accumulation (LARCA) and as actual (net) rate of C accumulation (ARCA) (see Fig. 2). LARCA is the ratio of total cumulative C and the peat column's basal age. ARCA is the current rate (i.e. the most recent 30 years) at which peatland
is sequestering C (Clymo et al., 1998). We also calculated near future rate of C accumulation (NFRCA) from 2001 to 2100 for the 10 studied zones (see below).



### 2.3.2 Climate change experiments

To investigate the sensitivity of CAR to climate change, the future experiments were performed (see Table 2) by extending the base experiment (BAS) covering the Holocene and recent past climate (to year 2000) for an additional century to year 2100 (Table 2). Climate output from the Coupled Model Intercomparison Project Phase 5 (CMIP5) RCP8.5 (Moss et al., 2010) runs performed with the Hadley Global Environment Model 2 (HadGEM2-ES) (Collins et al., 2011) was used to provide anomalies for 230 future climate forcing. HadGEM2-ES is an updated version of the same model chosen for the Holocene anomaly fields. It is in the middle-of-the-range of models contributing to the CMIP5 ensemble in terms of simulated temperature change across the Arctic region (Andrews et al., 2012; Klein et al., 2014). Model input of atmospheric $CO_2$ concentrations was taken from the RCP8.5 scenario, extracted from the International Institute for Applied Systems Analysis website (IIASA; 235 http://tntcat.iiasa.ac.at/RcpDb/; page visited 2 Feb 2017). In the first three experiments, the single factor effect of temperature (T8.5), precipitation (P8.5) and $CO_2$ (C8.5) was examined, followed by a combined experiment (FTPC8.5) where change in all three drivers was used to force the model. The model output variables examined here include total CAR, net primary productivity (NPP), net ecosystem C exchange (NEE), permafrost distribution, active layer depth (ALD) and regional soil C 240 balance.

### 2.3.3 Model evaluation

To evaluate the model, we compared the estimates of CAR with regional Holocene C accumulation 245 records synthesised across the pan-Arctic region, hereinafter referred to as "literature range". Second, time series were evaluated using observed Holocene LARCA values based on the 127 sites analysed by Loisel et al. (2014) and the 33 sites analysed by Yu et al. (2009) were averaged in 1000-year periods which we compared with our model simulations. The Loisel et al. (2014) dataset is more comprehensive and contains more basal points compared to Yu et al. (2009). In Yu et al. (2009), many key regions such 250 as the Hudson Bay Lowlands, western Europe, and western and eastern Siberia were not present, while the Loisel et al. (2014) dataset omits some regions such as Eastern Siberia and European Russia. Furthermore, the points in these two datasets were limited to 69 °N (< 69).

## 3 Results

### 3.1 Hindcast experiment

While peatland initiation started at ca. 12-13 kyr BP in high latitude areas, the majority of peatlands formed after 10 kyr BP (See Fig. A1). Mean modelled CAR among all 180 sites was 35.9 g C m$^{-2}$ y$^{-1}$,




after which it followed a similar temporal pattern to observed CAR values (Fig. 3a) (Yu et al., 2009;
      Loisel et al., 2014). The observed rate calculated by Yu et al. (2009) shows a dip after 5 kyr BP but the
      modelled result exhibited no such deviation (Fig. 3a). The observed rate reported by Loisel et al. (2014)
      is a little higher than the simulated rate before 4 kyr BP and for the present climate. Modelled CAR was
      higher at the beginning of the simulation in all the zones apart from Zone J (Fig. 3b). Zones A and B,
covering the Scandinavian and European regions, had high CAR in the beginning of the Holocene,
      which then declined through the Holocene phase, while Zone E covering eastern Siberia displays a peak
      suggesting an accelerated rate of C accumulation by the year 1900. Almost all regions exhibited similar
      CAR for 7-8 kyr BP, following different trajectories thereafter.

Scandinavia (Zone A), Europe (B), southwest Siberia (D), central Canada (H) and northern Canada (J)
      exhibit lower LARCA values compared to the pan-Arctic average (Fig. 4I) with north Canadian (J) and
      European (B) sites accumulating the lowest amounts (14.5±14.8 and 14.2±3.7 g C m$^{-2}$ yr$^{-1}$ respectively)
      of C through the Holocene. The other five zones (C, E, F, G and I) showed relatively higher mean
      LARCA values and the peatlands in eastern Siberia (E), Alaska (F) and western Canada (G) had the
highest mean LARCA values (26.8 ± 13.8, 26.4 ± 16.3 and 26.6 ± 14.7 g C m$^{-2}$ yr$^{-1}$ respectively). The
      global mean LARCA (black dotted line) for the 10 zones was 20.8 ± 12.3 g C m$^{-2}$ yr$^{-1}$ (Fig. 4I and Table
      1).

      Comparing mean ARCA for each zone with the respective LARCA values indicates that the majority of
sites accumulated relatively more C in the last 30 years except Scandinavia (A), and in Europe (B) the
      changes were almost negligible (Fig. 4II). The global mean ARCA for the last 30 years was 29.4 ± 27.8
      g C m$^{-2}$ yr$^{-1}$ suggesting an upward trend in CARs since the beginning of the Holocene (Fig. 4II and
      Table 1).

Interpolated values of permafrost (characterised in this study by ice fraction in the peat soil), ALD,
      CAR and accumulated litter for the month of September are presented for the recent past and future
      climate in Fig. 5, 6, 7 and A2. Figure 5(a) shows that permafrost was widely distributed from Siberia to
      Canada and in parts of northern Scandinavia around the end of 20$^{th}$ century according to our model. The
      majority of these permafrost areas were associated with shallow active layers (ALD < 0.1 m), while in
the southern parts of Siberia and Canada the active layers relatively deeper (Fig. 6a). The presence of
      permafrost is associated with diverse levels of peatland CAR (Fig. 7a), ranging from moderate to high
      litter accumulation (Fig. A2a). Large parts of western Canada, Alaska and Siberia accumulated
      relatively high amounts of C by the year 2000 (Fig. A2a) according to our model.



### 3.2 Climate change experiment

In the FTPC8.5 experiment, where all the drivers were combined,, a marginal decrease in global mean FLARCA (20.78 g C m$^{-2}$ yr$^{-1}$) compared with the mean LARCA (20.8 g C m$^{-2}$ yr$^{-1}$) (see Fig. 2) was noticed. However, the change in CAR was quite evident in certain geographic zones (Fig. 8I and bars). Some regions showed an increase in their C sequestration capacity while others become C neutral or sources of C. While Scandinavian (A), European (B), central and eastern Canadian (H, I) sites are projected to become C sources (Fig. 8I and bars), the remaining zones are projected to enhance their C sink capacity in this scenario. For example, the uptake capacity of northern Canadian (J) sites is projected to increase fourfold, to 52.3 ± 37.0 g C m$^{-2}$ y$^{-1}$, from (its LARCA value of) 14.5 ± 14.8 g C m$^{-2}$ y$^{-1}$ (Table 1 and Fig. 8). All zones showed a decline in CAR in T8.5 experiment, relative to the recent historical climate (Fig. 8II). This is explained by the positive effects of temperature on soil organic matter decomposition rates. An exception to this general pattern is seen for northern Canada (Zone J) sites where warming has a positive effect on CAR (Fig. 8II and bars; positive bar value means C source): higher temperatures create a more suitable environment for plant growth in this region where cold weather and permafrost limit plant (and therefore litter) production under present climate conditions (see Fig. A3). The mean modelled global NFRCA in the T8.5 experiment from the year 2000 to 2100 was 1.52 g C m$^{-2}$ yr$^{-1}$. This was a significant drop when compared to modelled LARCA and ARCA. In this experiment, the ESM-derived (Collins et al., 2011) surface air temperature anomalies used to force our model increase by approximately 5°C by 2100 relative to 2000. Higher temperature is associated with elevated decomposition rates leading to more C loss and higher heterotrophic respiration. Projected precipitation increases in the P8.5 experiment resulted in higher CAR in all zones. Regionally, Siberia and Far East Russia (C, D, E), Alaskan (F) and Canadian (G, H, I, J) sites showed the largest changes, while very little change was seen for the Scandinavia (A) and European (B) sites (Fig. 8III and bars). Elevated CO$_2$ in the atmosphere enhanced the rate of plant photosynthesis which led to higher CAR in the C8.5 experiment in all zones (Fig. 8IV and bars).

Our simulations suggest that the significant temperature increase implied by the RCP8.5 future scenario will lead to disappearance or fragmentation of permafrost from the peat soil, and deeper active layers, (Fig. 5b). The resultant change in soil water levels and saturation could then either suppress of accelerate the decomposition rate at many peatland locations (Fig. 8II). In the Siberian (C, D and E) and Alaskan (F) zones, the projected higher decomposition rates are compensated by higher plant productivity (Fig. 8III and IV); bars), leading to a net increase in CAR by 2100 in this scenario.

From Figure 5(b), it is evident that permafrost area declines, remaining limited to central and eastern parts of Siberia and the north Canadian region under the future experiment in our model (FTPC8.5). Permafrost disappears from the large parts of western Siberia and southern parts of Canada with very





little remaining presence in Scandinavia (Fig. 5b). This degradation in permafrost leads to deeper active
layers (Fig. 6b,c) and wetter conditions initially in large areas of peatlands currently underlain by
permafrost. Wetter conditions together with $CO_2$ fertilization lead to high CAR in these areas with high
C build up. In contrast, non-permafrost peatlands showed a decline in CAR and in total litter
accumulation due to higher decomposition rates (Figs. 7b,c and A2b,c) as a result of increases in
evapotranspiration, which draw down the WTP (not shown here).

**4 Discussion**

LARCA expresses the rate of C accumulated in a peatland since its inception (Clymo et al., 1998) and is
a useful metric of the sequestration capacity of peatlands because the current C uptake rate (ARCA) is a
snapshot in time that is not expected to reflect the C balance dynamics through the history of the
peatland (Lafleur et al., 2001; Roulet et al., 2007). Recent CAR tends to be higher compared to LARCA
because older peat would have experienced more decay losses, leaching and erosion (Lafleur et al.,
2001). This is clearly reflected in our result (Table 1) where LARCA < ARCA in most cases, even
though in our study only decay losses were considered.

The variability in LARCA among sites within a region with relatively similar climate highlights the
influence of local factors (Borren et al., 2004). If climate was the major driving factor behind observed
variations in LARCA then all the peatland types within one climate zone would be expected to have
similar LARCA values. LARCA is highly influenced by local hydrology, topography, climate
conditions, permafrost, fire events, substrate, microtopography and vegetation succession (Clymo,
1984; Robinson and Moore, 2000; Beilman, 2001; Turunen et al., 2002; Turetsky et al., 2007).
Furthermore, some studies attribute differences in LARCA values to the overrepresentation of
terrestrialised peatlands and an underrepresentation of paludified or shallow peatlands (Botch et al.,
1995; Tolonen and Turunen, 1996; Clymo et al., 1998) in estimations of this metric. Our model
initialisation allowed vegetation to reach an equilibrium with the climate of 10 kyr ago, but the model
ignores the presence of ice over some parts of the study area at this time, thus overestimating the
vegetation cover at the beginning of the simulation, leading to higher CAR than observed (Fig. 3a,b). In
addition, the underlying topography is a major factor for peat initiation and lateral expansion of any
peatland complex but no such data are available for regional simulations. Therefore, we assumed a
moist, on average horizontal, soil surface upon which peatland could potentially form at each of our 180
simulation points, ignoring the role of underlying topography and its effects on water movement within
a basin. However, the lateral exchange between higher and lower patches within an overall horizontal
landscape was included in our model (see Section 2).





The mean modelled LARCA across the pan-Arctic study area was $20.8 \pm 12.3$ g C m$^{-2}$ yr$^{-1}$, a value that
falls within the reported range for northern peatlands, namely 18.6-22.9 g C m$^{-2}$ yr$^{-1}$ (Yu et al., 2009;
Loisel et al., 2014). However, the Loisel et al. (2014) dataset is not completely representative of the
pan-Arctic region and data from some key regions are missing, such as eastern Siberia and European
Russia (Yu et al., 2014a). The Loisel et al. (2014) dataset includes points that are mainly from
deep/central parts and shallow peat basins are underrepresented (MacDonald et al., 2006; Gorham et al.,
2007; Korhola et al., 2010). Furthermore, the dataset is limited to areas north of 69 °N. Inclusion of
shallow peatland complexes and more sub-arctic and arctic sites in the synthesis might conceivable
bring down the mean observed LARCA value. Nevertheless, the overall trend of the modelled, global-
averaged CAR (n = 180) for the last 10 kyr is quite similar to these published syntheses (Fig. 3a and b
and Table 1).

Suitable climate (moist and cool) and optimal local hydrological conditions influenced by favourable
underlying topographical settings accelerated the CAR which led to the formation of large peatland
complexes in the pan-Arctic region (Yu et al., 2009). CAR is the balance between biological inputs
(litter accumulation) and outputs (decomposition and leaching) and these two important processes are
quite sensitive to climate variability (Clymo, 1991). High CAR is associated with high plant
productivity and a moist climate leading to shorter residence time in acrotelm layers and generation of
recalcitrant peat, or a combination of any of these factors (Yu, 2006). In many regions, CAR is also
influenced by the presence of permafrost. Under stable or continuous permafrost conditions, the CAR
slows down or ceases (Zoltai, 1995; Blyakharchuk and Sulerzhitsky, 1999) due to low plant
productivity. CAR may also become negative due to wind abrasion and thermokast erosion, but these
factors are not considered in our simulations. In contrast, areas underlain by sporadic and discontinuous
permafrost sequester relatively more C (Kuhry and Turunen, 2006).

Significant increases in temperature are expected at high latitudes in the coming century, even under the
most optimistic emissions reduction scenarios. Under these conditions, some peatlands could sequester
more C (Charman et al., 2013) while others could turn into C sources and degrade (Ise et al., 2008; Fan
et al., 2013). Permafrost peatlands are sensitive ecosystems and respond quite rapidly to temperature
change as well as other aspects of climate (Christensen et al., 2004). The formation of thermokast lakes,
degradation of palsa, flooding and subsidence of the land surface are key features that might be
indicators of and the result of rapid warming and permafrost decay. Soil subsidence-driven pond
formation has been observed to lead to a total shift from a recalcitrant moss-dominated vegetation
community to dominance by non-peat forming plant types such as *Carex* sp. (Malmer et al., 2005).
However, the complex physical process inducing such changes is not included in this model.



In our scenario simulations (Table 2), we find that higher temperature leads to thawing of permafrost
that in turn increases the moisture availability, at least initially. The rise in temperature also results in
early spring snowmelt and a longer growing season (Euskirchen et al., 2006) while, in the same time
frame, atmospheric $CO_2$ concentration will also increase. These factors lead to increases in plant
productivity, leading to higher CAR (Klein et al., 2013), even in cases where moisture- and
temperature-driven peat decomposition also speeds up.

High temperature and limited moisture conditions with limited or no permafrost are associated with an
increase in the Bowen ratio, implying a shift towards drier conditions which accelerates the peat
decomposition and in turn reduces CAR (Franzén, 2006; Ise et al., 2008; Bragazza et al., 2016). This
will also result in draw down of water position and dominance of woody shrubs. The latter trend,
namely an expansion of shrubs across the Arctic and beyond in the next half of the 21st century, is in
keeping with other studies (Sturm et al., 2005; Loranty and Goetz, 2012). Conversely, warmer and
wetter future climate conditions, in combination with $CO_2$ fertilization, could lead to increased CAR in
areas projected to have a higher precipitation rate, compensating temperature enhancement of
decomposition.

Simulated responses of peatland to the differential climate conditions of the studied regions reflect a
range of model responses, and are discussed in relation to available literature knowledge below.

**4.1 Scandinavia and Europe (Zones A and B)**

In the Scandinavian region (Zone A), observed mean LARCA values vary between 11.8 and 26.1 g C
$m^{-2}$ $y^{-1}$ (Tolonen and Turunen, 1996; Makila, 1997; Clymo et al., 1998; Makila et al., 2001; Makila and
Moisanen, 2007). Modelled averaged LARCA for this zone (17.2 ± 7.4 g C $m^{-2}$ $y^{-1}$) is within the
reported literature range (Fig. 4 Zone A and Table 3). A more representative LARCA estimate derived
from 1302 dated peat cores from all Finnish undrained peatlands is 18.5 g C $m^{-2}$ $y^{-1}$ (Turunen et al.,
2002), which is also quite close to our estimate. LARCA estimates from 10 sites in northern Sweden
range from 8-32 g C $m^{-2}$ $y^{-1}$, with an average of 16 g C $m^{-2}$ $y^{-1}$ (Klarqvist et al., 2001a). Estimates of
LARCA from Russian Karelia, adjacent to Scandinavia, are reported as 20 g C $m^{-2}$ $yr^{-1}$ (Elina et al.,
1984). The recent observed rate (ARCA) ranges between 8.1-23.0 g C $m^{-2}$ $yr^{-1}$ (mean 12.1 g C $m^{-2}$ $y^{-1}$)
for Scandinavia (Korhola et al., 1995) which can be compared to the modelled ARCA value (13.6 ±
18.2 g C $m^{-2}$ $y^{-1}$) in this zone.

The modelled LARCA (14.2 ± 3.7 g C $m^{-2}$ $y^{-1}$) for central and eastern Europe (Zone B) is relatively low.
However, while some sites in this region are reported as being quite productive (21.3 ± 3.7 g C $m^{-2}$ $y^{-1}$;
Anderson (2002)), long-term CAR estimates are available for relatively few sites (Charman, 1995;





Anderson, 1998), making a comparison difficult. The points that fall in the British Isles showed lower modelled LARCA (12-14 g C m$^{-2}$ y$^{-1}$) values than observed literature range indicating shortcomings in the simulation of local hydrological conditions, or a possible bias in the climate forcing of our model. A decline in CAR in Scandinavia and Europe over recent decades is apparent in our simulations. Some
observational studies also point to a reduced rate of C accumulation in recent years for this region (Clymo et al., 1998; Klarqvist et al., 2001b; Gorham et al., 2003). This slowing has been attributed to an increase in decay rates due to climate and hydrological changes, the development of a stable structure (Malmer and Wallen, 1999), divergence in the rate of nutrient supply, or a combination of these factors (Franzén, 2006). Our model predicts that the C sequestration capacity of the Scandinavian region will
decrease after 2050 and become C neutral, with peatland in the European region becoming a C source in the same time frame (Fig. 8(I) Zones A and B). The simulated future C losses are associated with increase in the decomposition rate due to higher temperatures and a lower soil water table, the latter resulting from the combination of marginal or no increase in precipitation and soil water loss due to higher evapotranspiration.

**4.2 Siberia (Zones C and D) and Far East Russia (Zone E)**

Large peatland complexes were formed in western Siberia during the Holocene and around 40% of the world's peat deposits are found in this region, covering more than 300 million ha (Turunen et al., 2001; Bleuten et al., 2006). LARCA for west Siberia has been estimated at 5.4 to 38.1 g C m$^{-2}$ y$^{-1}$ (Beilman et
al., 2009). The modelled LARCA for the northwest and southwest region is 24.6 ± 14.6 and 16.7 ± 8.6 respectively (Fig. 5 Zones C and D and Table 3). The combined average modelled LARCA for the north and southwest Siberian (C+D) zones is 20.6 g C m$^{-2}$ y$^{-1}$. Turunen et al. (2001) report average LARCA from 11 sites in northwestern Siberia at 17.2 g C m$^{-2}$ yr$^{-1}$ (range from 12.1 to 23.7 g C m$^{-2}$ yr$^{-1}$). Botch (1995) estimated relatively higher LARCA (31.4-38.1 g C m$^{-2}$ y$^{-1}$) for the raised string bogs in western
Siberia. These observations are in line with our modelled range of 24.6 ± 14.6 g C m$^{-2}$ yr$^{-1}$ for the northwestern sites.

Borren et al. (2004) found LARCA values between 19-69 g C m$^{-2}$ y$^{-1}$ for the southern taiga zones of southwestern Siberia. The modelled LARCA value for the southwestern zone (D) is 16.7 ± 8.6 g C m$^{-2}$
y$^{-1}$. The apparent underestimation by our model could be explained by the relatively larger area encompassed by our simulations, extending into warmer southerly areas with limited peat accumulation, compared to the aforementioned studies (Fig. 5 Zones C and D and Table 3). Borren and Bleuten (2006) modelled a LARCA range of 10-85 g C m$^{-2}$ y$^{-1}$ (mean 16 g C m$^{-2}$ y$^{-1}$) for a large mire complex in southwestern Siberia and our value falls within this range.





The mean observed LARCA was $10.6 \pm 5.5$ g C m$^{-2}$ y$^{-1}$ for a permafrost polygon peatland of Far East Russia (Gao and Couwenberg, 2015). Botch et al. (1995) cite CAR values of 44.8 gC m$^{-2}$ y$^{-1}$ for both Kamchatka and Sakhalin regions and 33.6 g C m$^{-2}$ y$^{-1}$ for Far East regions. Our modelled estimate 26.8 $\pm$ 13.8 g C m$^{-2}$ y$^{-1}$ is broadly comparable to the range of these observations.

Our model predicted that the sink capacity (22.7 g C m$^{-2}$ y$^{-1}$) of the entire Russian region (C, D and E) was higher than the pan-Arctic average (Fig. 4 and Table 3). In future, higher temperature and precipitation, together with increases in snow depth, result in permafrost degradation that will lead to a deeper active layer in the western part of Siberia (Fig. 5b, 6b). Plants experience improved hydrological

conditions due to a deeper ALD. Thawing of the permafrost in the peat and mineral soils coupled with a longer growing season and $CO_2$ fertilization leads to higher plant productivity, offsetting the higher decomposition rate leading to an increase in CAR (Fig. 7b, c). Hence, this region is projected to act as a C sink in the future (Fig. 8 a). It is notable in our simulations that temperature increases in the T8.5 experiment have a very limited overall effect on decomposition rate in Russia (Zones C, D and E) while

precipitation and $CO_2$ fertilization have a positive effect on C build up (Fig. 8 b, c and d).

### 4.3 Canada (zones G to J) and Alaska (zone F)

Canada's Mackenzie River Basin and Hudson Bay Lowlands are two of the largest peatland basins in

the world (Beilman et al., 2008). The individual observed C accumulation rates vary considerably across Canada and the LARCA for the entire Canadian region ranges from 0.2 to 45 g C m$^{-2}$ y$^{-1}$ (see Table 3). The modelled mean LARCA value averaged among zones G-J (the entire Canadian region) is 21.2 g C m$^{-2}$ y$^{-1}$. Most observational studies have been carried out in the western and central regions of Canada (Halsey et al., 1998; Vitt et al., 2000; Beilman, 2001; Yu et al., 2003; Sannel and Kuhry, 2009).

However, in recent years, studies have been conducted in the Hudson Bay and James Bay Lowlands of eastern Canada (Loisel and Garneau, 2010; van Bellen et al., 2011; Bunbury et al., 2012; Lamarre et al., 2012; Garneau et al., 2014; Holmquist and MacDonald, 2014; Packalen and Finkelstein, 2014). Observed LARCA in zone I is relatively low, as peatlands initiated later in this region due to a late Holocene thermal maximum (5.0–3.0 kyr; Yu et al., 2009) and the presence of the remnants of the

Laurentide ice sheet (Gorham et al., 2007). In our model simulations, all peatlands were initiated at the same time and we have not considered the influence of ice sheet cover, explaining the higher modelled CARs (25.3 $\pm$11.8 g C m$^{-2}$ y$^{-1}$) in the eastern region. The observed LARCA of the three main eastern regions in Canada is: Quebec (26.1 g C m$^{-2}$ y$^{-1}$; Garneau et al., 2014), Hudson Bay Lowlands (18.5 g C m$^{-2}$ y$^{-1}$; Packalen and Finkelstein, 2014) and James Bay Lowlands (23.9 g C m$^{-2}$ y$^{-1}$; Holmquist and

MacDonald, 2014), and other studies in the area have similar values (see Table 3). Our simulations suggest that permafrost will disappear from large areas of southern Canada under the RCP8.5 climate change scenario, leading to deeper ALD (Fig. 5b, 6b). While western and northern Canadian regions



sequester C at higher rates from 2001 to 2100 in our simulations, central and eastern parts turn into a C source over the same time period. Decompositions rate will increase due to higher temperatures,

overriding the positive gains due to precipitation and C fertilization in central and eastern regions (Fig. 8 zones H and I).

The majority of simulated points in northern Canada (Zone J) are in the continuous or discontinuous permafrost region (Sannel and Kuhry, 2009). Observed LARCA values in this zone vary from 0.2 to

16.5 g C m$^{-2}$ y$^{-1}$ (see Table 3). Similarly, the modelled CAR of the northern Canadian sites was lowest (14.5 ±14.8 g C m$^{-2}$ y$^{-1}$) as a result of cold climate conditions (Table 4). The mean temperature in this zone is around -15 °C, with short growing seasons and low precipitation rates, the majority of which falls as snow. In some sites, negligible CARs were noticed due to extremely cold climate conditions that limited plant productivity. In other sub-arctic regions, similar effects of cold climate and permafrost

conditions have been observed. For instance, LARCA ranges from 12.5 to 16.5 g C m$^{-2}$ y$^{-1}$ for the center polygon peatlands in western Canada (Vardy et al., 2000) and 11 g C m$^{-2}$ y$^{-1}$ in the northern Yokon (Ovenden, 1990). Similarly, polygon peat plateaus in eastern Siberia have sequestered C at low rates (10.2 g C m$^{-2}$ y$^{-1}$) (Gao and Couwenberg, 2015). Lately, owing to recent climate warming and permafrost thaw, bioclimatic conditions have changed in these peatlands and many of them have seen

two to threefold increases in CAR (Ali et al., 2008; Loisel and Garneau, 2010), indicating a recent shift toward an increased C sink capacity. A fourfold increase in CAR, associated with permafrost thaw and increased primary productivity, was simulated under future warming by our model (Table 1 and Fig. 8 Zone J).

Alaska hosts around 40 million ha of peatland area (Kivinen and Pakarinen, 1981). Studies show that LARCA in this region ranges from 5 to 20 g C m$^{-2}$ yr$^{-1}$ (see Table 3). Our modelling results (26.4 ±16.3 gC m$^{-2}$ yr$^{-1}$) may be overestimations (Table 1 and Fig. 4 Zone F). The higher CAR values in our simulations are caused by high plant productivity, moist climate conditions, the generation of recalcitrant peat or a combination of these factors. This overestimation of CAR in Alaska casts doubt on

the simulated large future sink capacity of the study area (55.5 ±16.3 gC m$^{-2}$ yr$^{-1}$) under the RCP8.5 scenario.

### 4.4 Future climate impacts on peatlands

Our simulations under the strong RCP8.5 forcing scenario indicate a sharp reduction in the area underlain by permafrost in e.g. western Siberia and western Canada, leading to an initial increase in moisture conditions or wet surfaces there. The increase in moisture conditions can dampen the amplifying effects of temperature on decomposition rates, leading to net increase in CAR (Figs. 6, 7 and



8). By 2100, our model indicates that permafrost areas will be limited to eastern Siberia, northern and
western Canada and parts of Alaska (Fig. 7).

In future, areas currently devoid of permafrost, mainly Europe and Scandinavia, eastern parts of Canada
and European Russia, could lose a substantial amount of C due to drying of peat in conjunction with a
deeper WTP (Fig. 8). In a modelling study, (Ise et al., 2008) used a coupled physical–biogeochemical
soil model at a site in northern Manitoba, Canada, and found that peatlands could respond quickly to
warming, losing labile soil organic carbon during dry periods. Similarly, Borren and Bleuten (2006),
using a three-dimensional dynamic model with imposed artificial drainage to simulate the Bakchar bog
in western Siberia, indicated that LARCA will drop from 16.2 to 5.2 g C m$^{-2}$ y$^{-1}$ during the 21st century
due to higher decomposition linked to reduced peat moisture content. Our simulations are based on
climate forcing derived from the RCP8.5 scenario output from one earth system model (HadGEM2-ES).
We expect that simulated changes in permafrost and C accumulation would be more moderate and
slower if the model were forced with more moderate levels of climate change.

Overall, we found that Scandinavia, Europe, Russia and Central and eastern Canadian sites could turn
into C sources while the C sink capacity could be enhanced at other sites (Fig. 8). The greatest changes
were evident in eastern Siberia, northwest Canada and in Alaska. Peat production was initially
hampered by permafrost and low productivity due the cold climate in these regions but initial warming
coupled with a moisture rich environment and greater $CO_2$ levels could lead to rapid increases in CAR
by 2100 in this scenario. In contrast, sites that experience reduced precipitation rates and that are
currently without permafrost could lose more C in the future.

## 5 Conclusion

Our model, which uniquely among large-scale models of high-latitude ecosystems accounts for
feedbacks between hydrology, peat properties, permafrost dynamics and dynamic, climate-sensitive
vegetation composition, is able to reproduce broad, observed patterns of peatland C and permafrost
dynamics across the pan-Arctic region. Under a business-as-usual future climate scenario, we showed
that non-permafrost peatlands may be expected to become a C source due to soil moisture limitations,
while permafrost peatlands gain C in our simulations due to initial increase in soil moisture as a result
of permafrost thawing, which suppresses decomposition while enhancing plant production. We also
demonstrate that the extant permafrost area will be reduced and limited to central and eastern parts of
Siberia and the north Canadian region by the late 21st century, disappearing from large parts of western
Siberia and southern parts of Canada with very little presence in Scandinavia. Our modelling approach
contributes to our understanding of the long-term peatland dynamics at regional and global scale. The
implications of future climate change for peatland C stocks is also discussed. As such it complements



empirical research in this field but also requires its output for model development and evaluation. We
plan to incorporate methane biogeochemistry and nutrient dynamics in the next model update. In future,
the model will be coupled to the atmospheric component of a regional Earth system model to examine
the role of peatland-mediated biogeochemical and biophysical feedbacks to climate change in the Arctic
and globally.



## Acknowledgements

This study was funded by the Nordic Top Research Initiative DEFROST and contributes to the strategic research areas Modelling the Regional and Global Earth System (MERGE) and Biodiversity and Ecosystem Services in a Changing Climate (BECC). We also acknowledge support from the Lund University Centre for the study of Climate and Carbon Cycle (LUCCI). Simulations were performed on the Aurora resource of the Swedish National Infrastructure for Computing (SNIC) at the Lund University Centre for Scientific and Technical Computing (Lunarc), project no. 2016/1-441.



**Table 1.** Mean modeled C accumulation rates at different timescales in 10 zones

| Zone | Region | Latitude range ($\lambda$) | Longitude range ($\varphi$) | No. of points (n) | LARCA ($gCm^{-2} y^{-1}$) | ARCA ($gCm^{-2} y^{-1}$) | NFRCA (FTPC8.5) ($gCm^{-2} y^{-1}$) |
|---|---|---|---|---|---|---|---|
| A | Scandinavia | 50 to 75 | 0 to 30 | 20 | 17.2 ±7.4 | 13.6 ±18.2 | -5.2 ±18.4 |
| B | Europe | 45 to 75 | -10 to 60 | 20 | 14.2 ±3.7 | 14.2 ±14.6 | -28.1 ±28.5 |
| C | Northwest Siberia | 60 to 75 | 50 to 120 | 20 | 24.6 ±14.6 | 35.9 ±18.9 | 40.3 ±12.1 |
| D | Southwest Siberia and parts of central Asia | 45 to 60 | 50 to120 | 20 | 16.7 ±8.6 | 39.1 ±25.1 | 20.1 ±21.2 |
| E | Far east Russia and parts of central Asia | 45 to 75 | 120 to 180 | 20 | 26.8 ±13.8 | 50.7 ±43.6 | 42.1 ±23.5 |
| F | Alaska | 55 to 75 | 190 to 220 | 12 | 26.4 ±16.3 | 32.2 ±31.3 | 55.5 ±16.3 |
| G | Western Canada | 50 to 75 | 220 to 240 | 13 | 26.6 ±14.7 | 32.2 ±36.5 | 38.5 ±16.2 |
| H | Central Canada and parts of US | 45 to 60 | 240 to 270 | 20 | 18.3 ±7.9 | 24.8 ±12.2 | 3.1 ±21.0 |
| I | Eastern Canada and parts of US | 45 to 60 | 270 to 300 | 20 | 25.3 ±11.8 | 28.2 ±22.1 | -5.21 ±26.1 |
| J | Northern Canada | 60 to 75 | 240 to 300 | 15 | 14.5 ±14.8 | 23.7 ±28.9 | 52.3 ±19.2 |
| - | pan-Arctic | 45 to 75 | 0 to 360 | 180 | 20.8 ±12.3 | 29.4 ±27.8 | 18.3 ±47.2 |





**Table 2.** Summary of hindcast and global change experiments

| Experiment no. | Experiment name | Description of hindcast and future experiments |
|---|---|---|
| 1. | BAS | Base experiment |
| 2. | T8.5 | RCP8.5 temperature only |
| 3. | P8.5 | RCP8.5 precipitation only |
| 4. | C8.5 | RCP8.5 $CO_2$ only |
| 5. | FTPC8.5 | RCP8.5 including all treatments |





**Table 3.** Observed regional long-term rate of peatland C accumulation across northern latitude areas

| Individual zone | Country | Extent | Type | No. of cores (sites) | Climate zone | LARCA mean (range) $(gCm^{-2} y^{-1})$ | Reference |
|---|---|---|---|---|---|---|---|
| **Zone A and B** | **Scandinavia and Europe** | | | | | | |
| 1. | Finland | Entire | Bogs and fens | 1028 | Sub-arctic and boreal | 26.1 (2.8-88.6) | Tolonen and Turunen (1996) |
| 2. | Finland | Haukkasuo | Bogs | 79 | Boreal | 19.1 (16.7-22.3) | Makila (1997) |
| 3. | Finland | Entire | Bogs and fens | - | Sub-arctic and boreal | 21.0 | Clymo et al. (1998) |
| 4. | Sweden | North | Bogs and fens | 10 | Boreal | 16.0 (8-32) | Klarqvist et al. (2001a) |
| 5. | Finland | Entire | Bogs and fens | 1302 | Sub-arctic and boreal | 18.5 (16.9-20.8) | Turunen et al. (2002) |
| 6. | Finland | Luovuoma | Fen | 58 | Sub-arctic | 11.8 (5-30) | Makila and Moisanen (2007) |
| 7. | Finland | South and central | Bogs and Fens | 10 | Sub-arctic and boreal | 21.7 (19.4-24.0) | Makila (2011) |
| 8. | Scotland | North | Bogs | 3 | Boreal | 21.3 (11.5-35.2) | Anderson (2002) |
| **Zone C, D and E** | **Siberia and Far East Russia** | | | | | | |
| 1. | FSU[1] | Entire | Bogs and fens | - | Sub-arctic and boreal | 30 | Botch et al. (1995) |
| | Siberia | West | Bogs | - | Sub-arctic and boreal | 31.4 - 38.1 | Botch et al. (1995) |
| 2. | Siberia | Northwest | Bogs and fens | 11 | Boreal | 17.29 (12.1-23.7) | Turunen et al. (2001) |
| 3. | Siberia | Northwest | Bogs and fens | 23 | Sub-arctic | 17.1 (5.4-35.9) | Beilman et al. (2009) |
| 4. | Siberia | Southwest | Bogs and fens | 8 | Boreal | 19-69 | Borren et al. (2004) |
| 5. | Siberia | Kamchatka | Bogs | - | - | 44.8 | Botch et al. (1995) |
| 6. | Siberia | Sakhalin | Bogs | - | - | 44.8 | Botch et al. (1995) |
| 7. | Siberia | Far East region | Bogs | - | - | 33.6 | Botch et al. (1995) |

[1] FSU- Former Soviet Union




| | | | | | | | |
|---|---|---|---|---|---|---|---|
| 8. | Siberia | Yakutia | Polygon peatland | 4 | Sub-arctic | 10.6 (8.9-13.8) | Gao and Couwenberg (2015) |
| **Zone F and G** | **Western Canada and Alaska** | | | | | | |
| 1. | W. Canada | - | Bogs and fens | - | Arctic, sub-arctic and boreal | 19.4 | Vitt et al. (2000) |
| 2. | Alaska | South-central | Bogs and fens | 4 | Boreal | 15 (5-20) | Jones and Yu (2010) |
| 3. | Alaska | South-central | Bogs and fens | 4 | Boreal | $11.5^2$ | Loisel and Yu (2013) |
| 4. | Alaska | - | Bogs and fens | - | Sub-arctic and boreal | 12.6 (8.6-16.6) | Gorham (1991) |
| **Zone H and I** | **Central and Eastern Canada** | | | | | | |
| 1. | E. Canada | Hudson Bay Lowlands, Ontario | Bogs and fens | 17 | Sub-arctic | 18.5 (14-38) | Packalen and Finkelstein (2014) |
| 2. | E. Canada | Hudson Bay Lowlands, Ontario | Bogs and fens | 1 | Sub-arctic | 18.9 (8.1- 36.7) | Bunbury et al. (2012) |
| 3. | E. Canada | Hudson Bay Lowlands, Quebec | Bogs and fens | 2 | Sub-arctic | 24 (23.19-24.19) | Lamarre et al. (2012) |
| 4. | E. Canada | James Bay Lowlands, Quebec | Bog | 3 | Boreal | 16.2 (14.4-18.9) | van Bellen et al. (2011) |
| 5. | E. Canada | James Bay Lowlands, Quebec | Bogs and fens | 13 | Sub-arctic and boreal | 23.6 (17.6-38.5) | Gorham et al. (2003) |
| 6. | N. America and E. Canada | Maine, Newfoundland and Nova Scotia | Bogs | 3 | Boreal | 34.8 (28.5-45) | Charman et al. (2015) |
| 7. | E. Canada | New Brunswick, Quebec, Ontario, Prince Edward Island, Nova Scotia | Bogs | 15 | Sub-arctic and boreal | 19 (5.1-34.6) | Turunen et al. (2004) |
| 8. | C. Canada | Upper Pinto fen, Alberta | Fen | 1 | Boreal | 31.1 | Yu et al. (2003) |
| 9. | C. Canada | Goldeye Lake | Fen | 1 | Boreal | 25.5 (7.8-113) | Yu (2006) |

---

[2] CAR over past 4000 years





| 10. | C. Canada | Central | Bogs and fens | 14 | Sub-arctic and boreal | 24.8 (8-37.5) | Yu (2006) |
|---|---|---|---|---|---|---|---|
| 11. | C. Canada | Alberta | Fens | 4 | Boreal | 32.5 (21.4-44.2) | Yu et al. (2014b) |
| 12. | C. Canada | Mariana | Fen | | Boreal | 33.6 (7.0-70.6) | Nicholson and Vitt (1990) |
| 13. | E. Canada | Hudson Bay and James Bay Lowlands | Bogs | 8 | Sub-arctic and boreal | 23.95 (16.5-33.9) | Holmquist and MacDonald (2014) |
| 14. | E. Canada | James Bay Lowlands, Quebec | Bogs | 4 | Boreal | 22.5 (9.1-41.7) | Loisel and Garneau (2010) |
| 15. | E. Canada | Quebec | Bogs | 21 | Sub-arctic and boreal | 26.1 (10-70) | Garneau et al. (2014) |
| **Zone J** | | | **Northern Canada** | | | | |
| 1. | N. Canada | - | - | 22 | Sub-arctic | 0.2 -13.1 | Robinson and Moore (1999) |
| 2. | N. Canada | Nunavut, Northwest Territories | Polygon peatlands | 4 | Sub-arctic and low arctic | 14.1 (12.5-16.5) | Vardy et al. (2000) |
| 3. | N. Canada | Yukon | - | - | Sub-arctic | 11.0 | Ovenden (1990) |
| 4. | N. Canada | - | - | - | Sub-arctic | 9.0 | Tarnocai (1988) |
| 5. | N and C. Canada | Selwyn Lake and Ennadai Lake | Peat plateau | 2 | Sub-arctic | 12.5-12.7 | Sannel and Kuhry (2009) |
| 6. | N. Canada | Baffin Island | - | - | Arctic and sub-arctic | 0.2-2.4 | Schlesinger (1990) |





**Figures:**

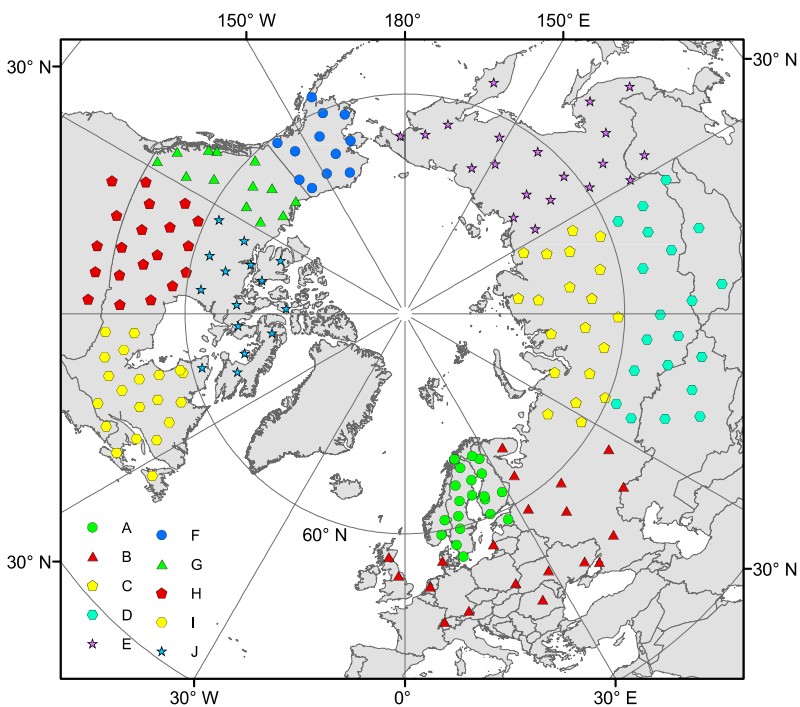

**Fig. 1** Location of 180 randomly selected simulation sites spread across 10 geographical zones between 45 and 75°N.





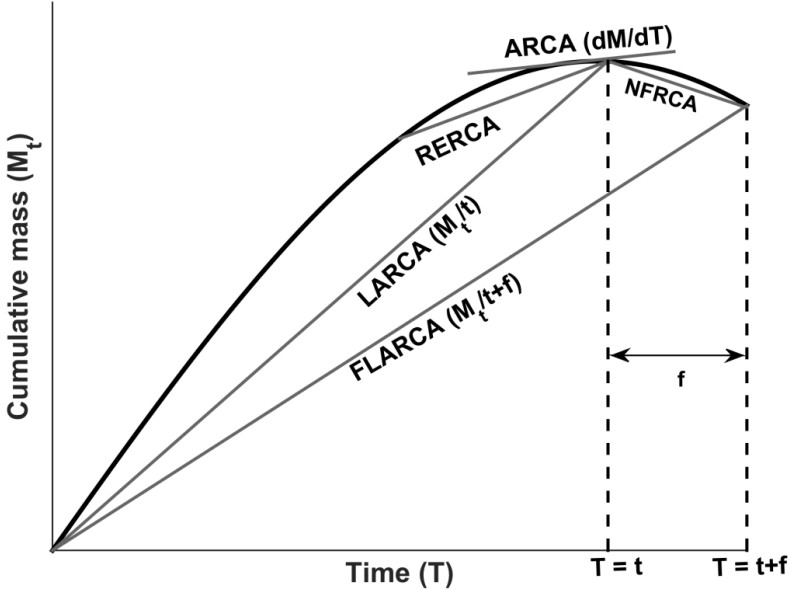

**Fig. 2** Commonly used measures of peat accumulation rate: long-term (apparent) rate of C accumulation (LARCA), recent rate of C accumulation (RERCA), actual rate of C accumulation (ARCA), simulated future long-term (apparent) rate of C accumulation (FLARCA), and near future rate of C accumulation rate (NFRCA) (Adapted from Rydin and Jeglum (2013))





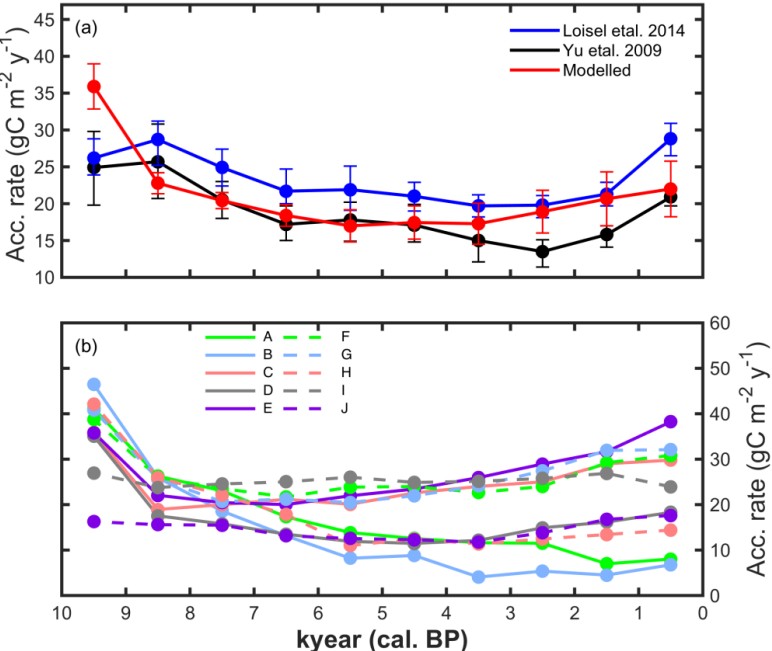

**Fig. 3** (**a**) Simulated and observed mean C accumulation rate (g C m$^{-2}$ y$^{-1}$) for each 1000-year period for the last 10,000 years. Red: simulated mean (and standard error of the means) CAR based on 180 random sites. Blue and black points observed C accumulation rates (g C m$^{-2}$ y$^{-1}$) based on 127 (Loisel et al., 2014) (blue points) and 33 sites (Yu et al., 2009) (black points) across northern peatlands with error bars showing standard errors of the means, (**b**) mean C accumulation rate (g C m$^{-2}$ y$^{-1}$) for each zone (Fig 1) for each 1000-year period for the last 10,000 years

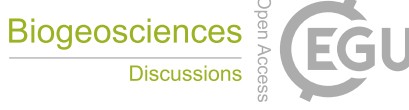



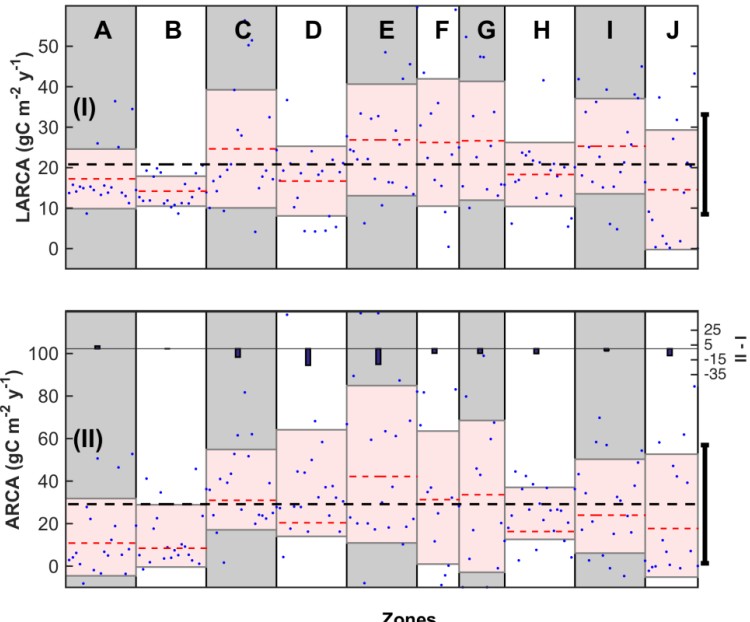

**Fig. 4** Simulated Holocene peat accumulation rates across the 10 zones considered in this study (blue dots) and for the pan-Arctic region as a whole (black lines). The x-axes shows the number of sites partitioned into 10 zones. The black dotted line is the global average with standard deviation (black line outside the y-axes) and the red dotted line is the average among zones with standard deviation in light red patch. (I) simulated long-term (apparent) rate of C accumulation (LARCA); (II) simulated actual rate of C accumulation (ARCA) for the last 30 years. Blue bars shows the difference between ARCA and LARCA mean values for the respective zone (II-I)




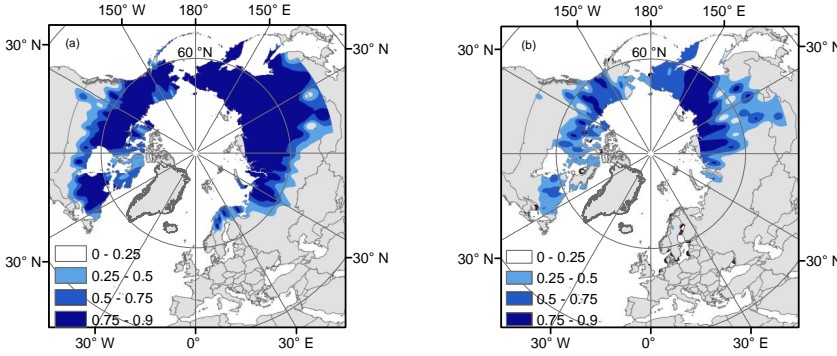

**Fig. 5** Modelled September ice fraction (0-1) in the peat soil (as a proxy for permafrost distribution) interpolated among simulation points, averaged over (a) 1990-2000 and (b) 2090-2100.




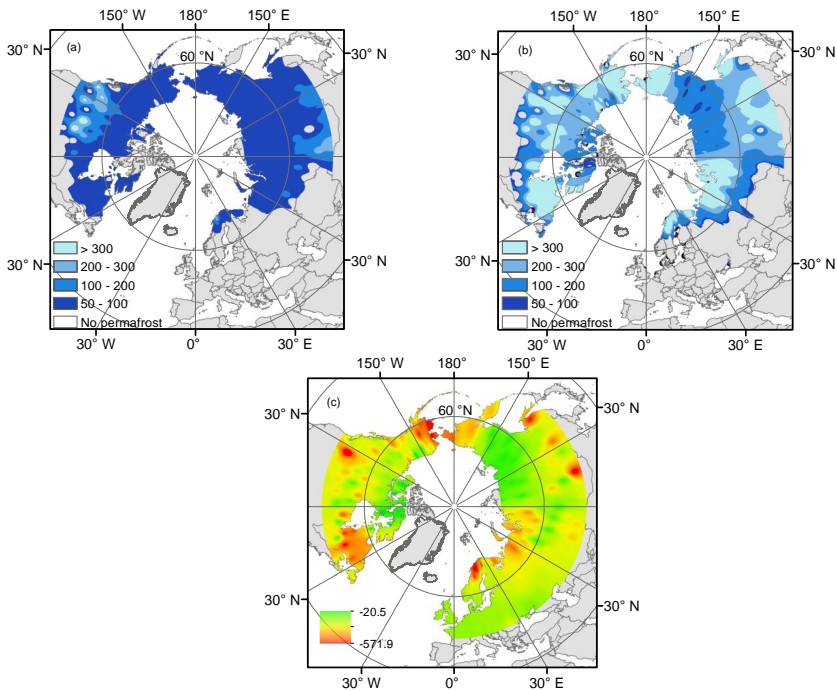

**Fig. 6** Modelled mean September active layer depth (ALD in cm) interpolated among simulation points for (a) 1990-2000, (b) 2090-2100; (c) Net change in total ALD (b-a).





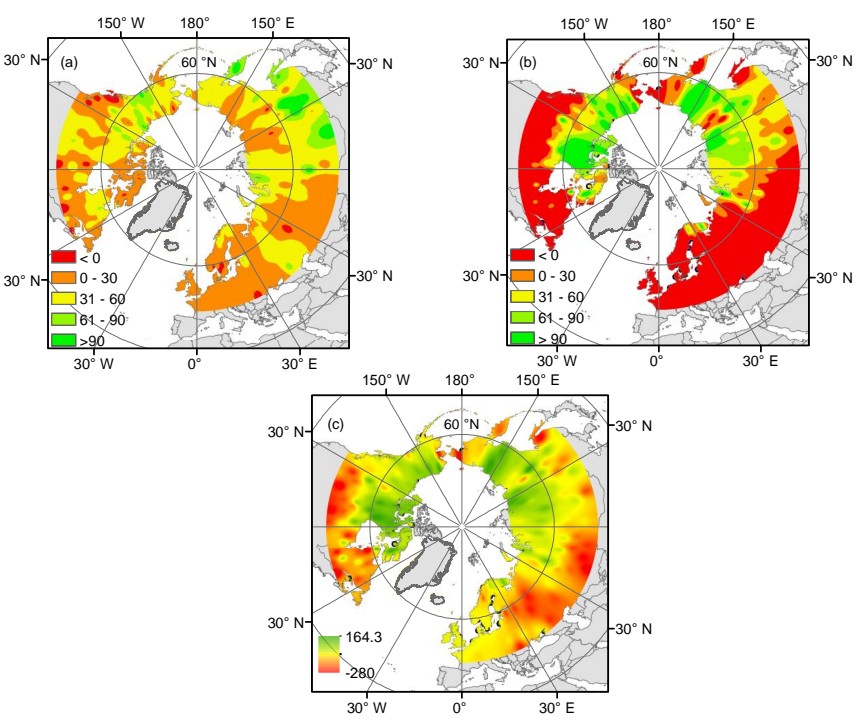

**Fig. 7** Modelled mean C accumulation rate (g C m$^{-2}$ y$^{-1}$) interpolated among simulation points for (a) 1990-2000, (b) 2090-2100; (c) Net change in total accumulation rate (b-a).



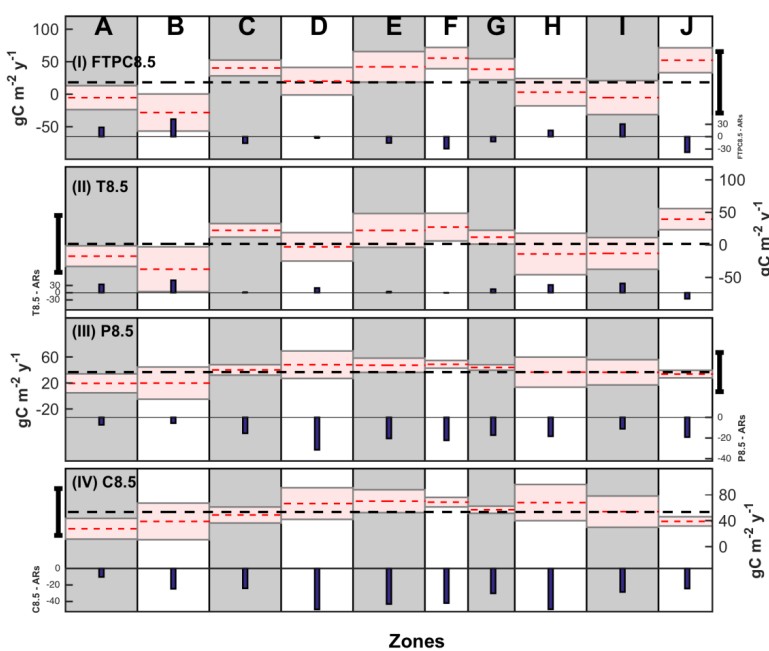

**Fig. 8** Simulated C accumulation rate (blue lines) for each zone (refer Fig. 1 and 4) and across the pan-Arctic. The black dotted line is the pan-Arctic average with standard deviation (black line outside); red dotted line is the average for the respective zone with standard deviation in light red patch (I) average simulated near future rate of C accumulation (NFRCA) from the year 2001-2100 in FTPC8.5 experiment; (II) simulated NFRCA in T8.5 experiment, (III) simulated NFRCA in P8.5 experiment and (IV) simulated NFRCA in C8.5 experiment. Blue bars show the difference between NFRCA and LARCA values for each zone.



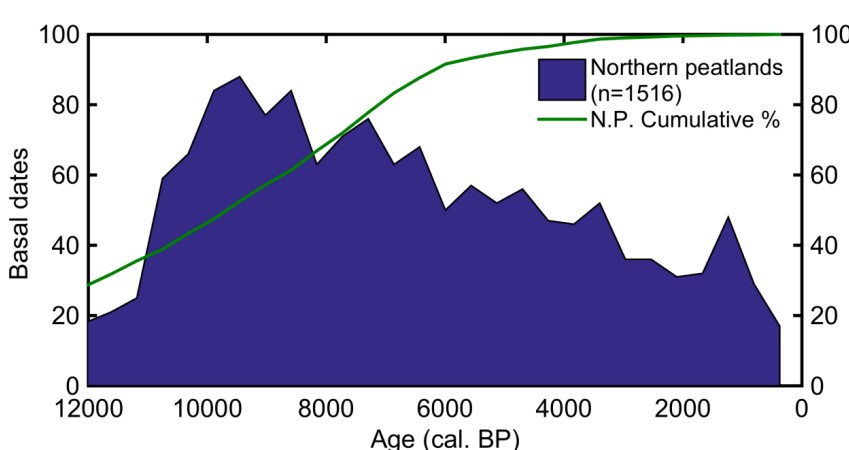

**Fig. A1** Observed peat basal ages plotted as a frequency curve and cumulative percentage (in green) for northern peatlands (n = 1516; MacDonald et al. (2006))





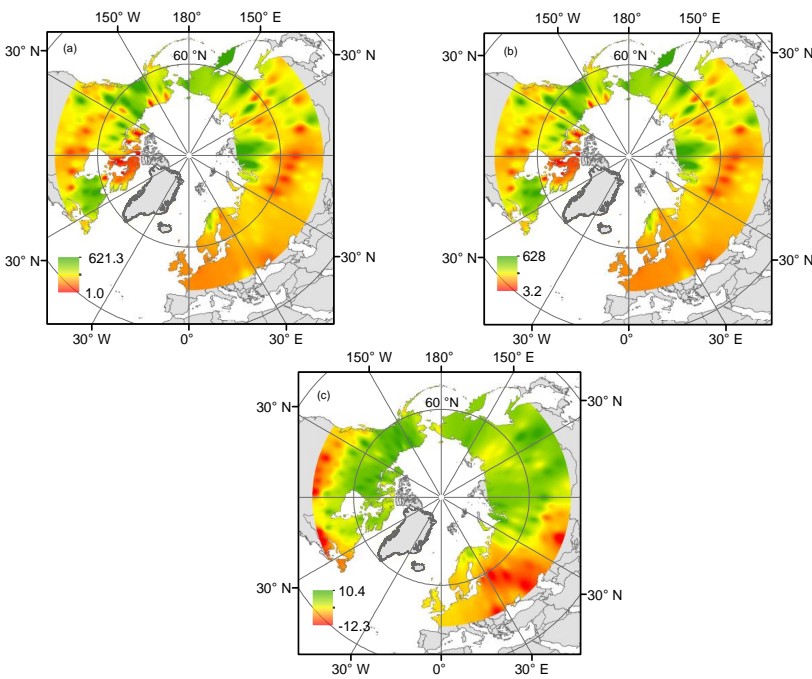

**Fig. A2** Modelled total accumulated C interpolated (kg C m$^{-2}$) among simulation points for (a) 1990-2000, (b) 2090-2100; (c) Net change in total C accumulation (b-a).





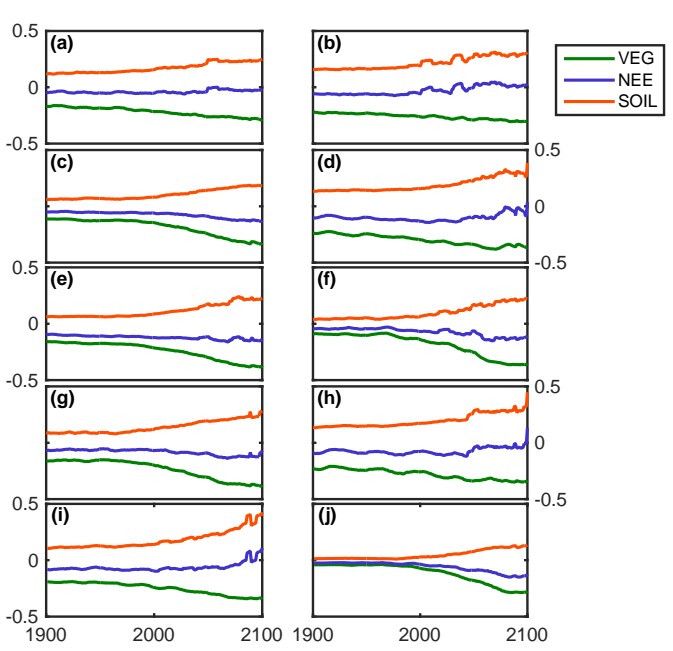

**Fig. A3** Total simulated carbon fluxes (10-year moving average) (in kg C m$^{-2}$ y$^{-1}$) for each zone for 1900-2100, including the RCP8.5 (FTPC8.5) forcing scenario for 2001-2100: vegetation NPP (VEG), litter and soil respiration (SOIL) and net ecosystem exchange (NEE).



**Table A1.** Plant functional types (PFTs) simulated in this study, showing representative taxa, phenology, bio-climatic limits, water table position (WTP) threshold for establishment, prescribed root fractions in mineral soil layers, and initial decomposition rate for different litter fractions.

| PFT (abbreviation) | Representative taxa | Phenology | Climate Zone | Growth Form | Min/Max temperature of the coldest month for establishment (ºC) | Max GDD for establishment (ºC day) | WTP threshold (in mm) | Root fraction Upper mineral soil (UM) | Root fraction Lower mineral soil (LM) | Litter fraction | Initial decomposition rate (k₀)[3] (yr⁻¹) |
|---|---|---|---|---|---|---|---|---|---|---|---|
| High summergreen shrub (HSS) | *Salix spp., Betula nana* | Summer green | Boreal-Temperate | Woody | -32.5/- | 1000 | < -250 | 0.7 | 0.3 | Wood | 0.055 |
| | | | | | | | | | | Leaf | 0.1 |
| | | | | | | | | | | Root | 0.1 |
| | | | | | | | | | | Seed | 0.1 |
| Low evergreen shrub (LSE) | *Vaccinium vitis-idaea, Andromeda polifolia* L. | Evergreen | Boreal-Temperate | Woody | -32.5/- | 100 | < -250 | 0.7 | 0.3 | Wood | 0.055 |
| | | | | | | | | | | Leaf | 0.1 |
| | | | | | | | | | | Root | 0.1 |
| | | | | | | | | | | Seed | 0.1 |
| Low summergreen shrub (LSS) | *Vaccinuim myrtillus, Vaccinium uliginosum, Betula nana* L. | Summer green | Boreal-Temperate | Woody | -32.5/- | 100 | < -250 | 0.7 | 0.3 | Wood | 0.055 |
| | | | | | | | | | | Leaf | 0.1 |
| | | | | | | | | | | Root | 0.1 |
| | | | | | | | | | | Seed | 0.1 |
| Graminoid (Gr) | *Carex rotundata* Wg., *Eriophorum vaginatum* L. | Evergreen | Boreal-Temperate | Herbaceous | -/- | - | > -100 | 0.9 | 0.1 | Leaf | 0.1 |
| | | | | | | | | | | Root | 0.1 |
| | | | | | | | | | | Seed | 0.1 |
| Moss (M) | *Sphagnum* spp. | Evergreen | Boreal-Temperate | Herbaceous | -/15.5 | - | < +50 and > -500 | - | - | Leaf | 0.055 |
| | | | | | | | | | | Seed | 0.055 |

---

[3] Aerts et al. (1999), Frolking et al. (2002) and Moore et al. (2007)





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

Zoltai, S. C.: Permafrost Distribution in Peatlands of West-Central Canada during the Holocene Warm Period 6000 Years Bp, Geogr Phys Quatern, 49, 45-541995.