# Peer review of "Modelling past, present and future peatland carbon accumulation across the pan-Arctic"

_Biogeosciences, 2017_

## Referee Comment (RC1) · Anonymous Referee #1 · 27 Mar 2017

In this manuscript, Chaudhary et al. present and discuss results from a simulation study of past and future carbon accumulation dynamics of pan-Arctic peatlands using a process-based model LPJ-GUESS. An earlier paper (available online as Biogeosciences Discussions paper) by the same authors presents details about revising this LPJ model version to simulate peatland C dynamics. In particular, I find that the inclusion of microtopographic features (patches of uneven height) and subsequent treatments of within-gridcell hydrology is novel and represents a major progress in simulating peatland carbon dynamics at large scale.

This manuscript uses this model to simulate peatland C dynamics at 180 grid cells across the pan-Arctic region, representing the major northern peatland regions. This is an interesting strategy to capture differences and similarities in spatial patters of C dynamics across the pan-Arctic region. This approach would not allow to quantify any

changes in total peat C stocks over time, as it does not explicitly prescribe or simulate peatland extent and its change over time. But still it is effective to evaluate the C accumulation rates at different sites/regions.

The simulated peat accumulation rates over the last 10 kyr are used to provide baseline information in evaluating future change in peat C accumulation rates. The simulation results show that peatlands in southern regions will become C sources due to moisture limitations in a warm climate, while peatlands further north will likely continue to be a C sink due to warming-induced increase in production. These results appear to be robust and will make an important contribution to our understanding of peatland C dynamics in a changing climate.

I think that this manuscript presents an important study and should be published after considering and addressing the following comments.

General comments: 1. I'd suggest that the authors briefly describe how they generate Holocene climate input data to drive the model. I understand that has been described in detail in the previous model description/calibration paper, but it would be useful to provide a concise description in the paragraph (Lines 191-206) as well (such as model-simulated paleoclimate).

2. I have some difficulties with those C accumulation terms as described in lines 217-221 and in Figure 2. I am familiar with but personally don't really like those acronyms. I think some are more useful than other. I think that LARCA is useful as it also represents long-term (10 kyr in this case) actual/"true" average C accumulation rates – which is equivalent to the mean time-weighted C rate from individual cores or synthesis products as cited in the paper. LARCA also should be the same as overall net C balance as discussed in Yu (2011), due to the same peat mass (conservation) through the last 10 kyr. So LARCA can be used to compare with present and future C accumulation (sequestration) rates. The difference in apparent and actual/"true" C accumulation rates was also discussed in Spahni et al. (2013) and Frolking et al. (2014).

However, I find ARCA is problematic, as it is actually a metric for apparent C accumulation rates – that is, dM/dT (T = 30-year period), despite its name "actual (net) rate of C accumulation". The true C balance rate should consider the decomposition C loss of the entire peat profile during that 30-year period. Am I missing something? A "true" C accumulation rate considering decomposition of previous accumulated peat is needed to compare with past 10 kyr (LARCA) and future C rates.

I find that the difference between FLARCA and LARCA is a useful metric to quantify the average true C accumulation rates in the future, so that should be the metric showing in Table 1 and discussed. Both NFRCA and RERCA are apparent C accumulation rates and may not be as useful. I don't think RERCA has been discussed much in the manuscript. I suggest they can be removed from discussion.

3. In general, the manuscript is well written and clearly organized. However, I find some text in subsection 3.2 belong to Discussion, rather than Results section. For example, lines 315-322 and lines 336-339. Perhaps these can be moved to Discussion section.

Specific comments: Title: I suggest to change to "…across the pan-Arctic region", as stated in some places of the text. It should change throughout the text.

L22: change to "central and eastern Canada" (lower case)

L28: change to "higher CO2"?

L47: either "organic-rich" or "C-rich", but both together a bit awakward

Paragraph l 57-74: A nice paragraph to summarize peatland models. I'd suggest to concisely mention the following models as well: Spahni et al. (2013), Stocker et al. (2014) and Wang et al. (2016). The first two used an LPJ model version to simulate peatland C dynamics, while Wang et al. (2016) used a different ecosystem model (TEM).

L165: add "," after "depth"

L193: change to "from 45 to 75. . ."

L196: defining 0 BP as 1900 is unnecessary and potentially confusing, as conventionally present = 1950 AD. I suggest just to call it 10,000 years before 1900 AD.

L198: describe concisely how Holocene climate input data were generated and prepared. See my general comments above.

L202: are the $CO_2$ concentration simulated or measured from ice cores? If they are ice-core based measurements, then you could just interpolate ice-core data for your purpose, rather than from the data used in UM model, which likely used the ice-core data in the first place. Clarify.

L217: see my general comments about these acronyms.

L299-300: I don't think the difference between 20.78 and 20.8 should be discussed. Are they robust enough for discussion? It would be useful to present these differences for different zones in Table 1, instead NFRCA. Apparently some zones accumulate more C in the future than other zones, and differences cancel out.

L306: the value of 53.2+/-37.0 is different from the one (error term) in Table 1. Check this and other values.

L315-322: move to Discussion?

L336-339: move to Discussion section?

L348: ARCA is an apparent C accumulation rate, so comparing it to LARCA is not very meaningful. But if overall decomposition is considered, it would become "true" C rate. See general comment above.

L384: change to "litter addition"?

L425: subsections 4.1 – 4.3 provide nice summaries of empirical data and their comparison with simulated results.

L546: change to "permafrost, for example in western Siberia. . ."

L552 and l585: change to "in the future"

L600: Table 1: -suggest to modify ARCA by considering decomposition loss from all previous peat. That way, it becomes an actual/"true" C accumulation rate. As now, it is a metric for apparent C rate that does not reflect the C sequestration rate and may not be appropriate to compare with LARCA and future C rates. See my general comments.

-Replace NFRCA by (FLARCA minus LARCA)? (FLARCA minus LARCA) represents mean "true" C sequestration rate from 2001 to 2100. See my comments above.

-Note for the Zone J, NFRCA is 52.3 +/-19.2, but it was indicated as 52.3+/-37.0 in the text on line 306 (different error terms). Check this and other values.

Figure 2: -suggest to redefine ARCA by considering decomposition C loss of all previous peat, and delete PERCA and NFRCA. See general comments above. -For FLARCA: change to (Mt/(t+f))? (add an extra pairs of brackets)

Figure 3: change X-axis label to "Age (ka BP)"

Figure 4: redefine ARCA?

Figure 5: I find these results are exciting. -I wonder if it is useful to add a panel to show (and discuss) the observed permafrost distribution (with various categories of continuous, discontinuous, etc.). -if so, three panels should be on one row from left to right, panels a, b, c (the same for Fig. 6) -perhaps Figures 5 and 6 can be merged as one figure with 5 or 6 panels in two rows, as they are relevant results and easier to compare.

Figure 7: Again, these results are exciting. -What does the "simulated mean C accumulation rate" mean? Is it apparent C rate or actual/"true" C rate (net C balance) that considers decomposition of all previous peat? -maybe better to put 3 panels in one row and move them closer.

Suggested references: Frolking S, Tlabot J and Subin ZM (2014) Exploring the relationship between peatland net carbon balance and apparent carbon accumulation rate at century to millennial time scales. The Holocene 24: 1167-1173. Spahni, R., F. Joos, B. D. Stocker, M. Steinacher, and Z. C. Yu (2013), Transient simulations of the carbon and nitrogen dynamics in northern peatlands: From the Last Glacial Maximum to the 21st century, Clim. Past, 9(3), 1287–1308.
 Stocker, B. D., Spahni, R., and Joos, F.: DYPTOP: a cost- efficient TOPMODEL implementation to simulate sub-grid spatio-temporal dynamics of global wetlands and peatlands, Geosci. Model Dev., 7, 3089–3110, doi:10.5194/gmd-7-3089- 2014, 2014. Wang, S., Q. Zhuang, Z. Yu, S. Bridgham, and J. K. Keller (2016), Quantifying peat carbon accumulation in Alaska using a process-based biogeochemistry model, J. Geophys. Res. Biogeosci., 121, 2172–2185
 Yu, Z. C.: Holocene carbon flux histories of the world's peatlands: Global carbon-cycle implications, Holocene, 21, 761–774, 2011.

---

## Referee Comment (RC2) · Anonymous Referee #2 · 6 Apr 2017

Review of "Modelling past, present and future peatland carbon accumulation across the pan-Arctic" by Nitin Chaudhary, Paul A. Miller and Benjamin Smith.

In their manuscript "Modelling past, present and future peatland carbon accumulation across the pan-Arctic" Chaudhary present and discuss results of their model for peat accumulation and decay for the past and the near future. The authors present peat accumulation rates at randomly selected locations and discuss how these change under changing climatic conditions.

Overall the manuscript presents novel results and is of interest to a substantial number of readers of biogesciences. However the current version of the manuscript is very difficult to follow and lacks clarity. It therefore requires major revisions before publication.

Over large parts of the manuscript the reader is wondering what the authors want to tell him or her. This is especially severe in sections 3 and 4, where results are reported and discussed. The section 3.2, first paragraph is one example: Here the authors discuss results of climate change experiments. Reading the paragraph feels like a near endless list of carbon accumulation rates defined in slightly different ways and for different regions. It is not possible to list all occurrences of lacking clarity, therefore I suggest the authors carefully look at the manuscript and rewrite unclear sections.

In addition the conclusions section is extremely weak and vague in discussing the conclusions, while half of the section consists of an outlook that is out of place in the conclusions section.

Finally, the authors only scratch at the surface of the capabilities of their model. LPJ-GUESS should be able to determine changes in vegetation composition – however these are not discussed in the manuscript. Similarly, the authors lay claim to unique capabilities of their model (Conclusions section) – however the results of these unique capabilities are not actually discussed.

In addition there are a number of minor issues:

- The climate forcing used to drive the Holocene experiment is unclear, the reader needs to read Chaudhary et al. (2016) in order to understand how it was derived. A two sentence summary how it was derived, including the Miller et al reference, would help.

- Fig. A1 is a reproduction from MacDonald et al. 2006. It is therefore not needed, the authors can refer to the original figure.

- Page 8 / line 289: The authors refer to regions with shallow active layers (ALD < 0.1m) and refer to their Fig. 6a. This is impossible to follow, since the Fig. only show ALD 50-100 cm, 100-200, 200-300 and > 300 – the range referred to in the text is not shown.

- Page 9 / line 325: the authors refer to Fig. 5b. I assume they mean 6b?

- Page 9, lines 324-329: The paragraph deals with temperature effects. The authors then refer to their Fig. 8 III and IV – however panels III and IV show the precipitation and CO2 effects. Clearly there is some logical error in this paragraph.

- Page 11, line 375: The authors write that Loisel et al. is limited to north of 69°N. However it is south of 69°N that is meant.

- Page 11, lines 381 ff: unclear, when moist conditions played a role

- Page 12, lines 415-418: Is this trend only reported in the literature, or does it also occur in the model?

- Page 14, line 490: the authors refer to Fig. 8 b, c and d, but they mean II, III, and IV

- Fig. 4: What the authors call a "dotted" line in the Figure legend is usually referred to as a "dashed" line. In addition, the black line discussed in the legend is invisible in the Figure.

- Fig. 6: Choice of colours is less than perfect. 1) Are no active layer depths of less than 50cm shown? This is implied by Figures 6a and b. 2) the colour scale chosen in Fig. 6c usually implies a symmetric range from positive (green) to negative (red), with no change indicated by yellow. However in this Figure all values are negative.

- Fig. 7c: Colour scale not symmetric – zero value unclear (see also my comment to Fig. 6, part 2)

- Fig. 8: The same dashed / dotted issue as in Fig. 4

- Fig. A2: Colour scales are not centered around zero and are different between plots, making comparisons very hard.

- Fig. A3: On an A4 printout, this Figure is still too small to see any details. In addition, there are no axis subdivisions between -0.5/0/0.5, making it extremely hard to read

anything from the Figure.

---

## Author Comment (AC1) · 24 May 2017

We appreciate the time and effort spent by the editor and the reviewers in reviewing this manuscript. We have addressed all the issues indicated in the review reports.

General comments: 1. I'd suggest that the authors briefly describe how they generate Holocene climate input data to drive the model. I understand that has been described in detail in the previous model description/calibration paper, but it would be useful to provide a concise description in the paragraph (Lines 191-206 in the original manuscript (OM)) as well (such as model- simulated paleoclimate).

Response: We avoided a detailed description of the Holocene climate input data to make the paper more concise, referring instead to our companion Paper I (bg-2106-319) but based on the suggestions of the reviewers, we have now included a more

detailed description of the Holocene climate input data in this paper.

Revised Text: Each simulation was run for 10,100 years, and comprised three distinct climate-forcing periods. The first, Holocene, phase lasted from 10 kyr before present (BP) until 0 BP. During this period, the model was forced with daily climate fields (temperature, precipitation and cloudiness) constructed by interpolating between monthly values from the year 10,000 calendar years before present (cal. BP) until the year 1900. The monthly Holocene climate forcing data was prepared by the delta-change method by applying the relative monthly anomalies of temperature and precipitation the nearest GCM gridcell to the site location to their average monthly values from the CRU TS 3.0 global gridded climate data set (Mitchell and Jones, 2005) from the period 1901 to 1930. We then linearly interpolated the values between the millennium time slices to get values for each year of the simulation. This method conserves the inter-annual variability for temperature and precipitation throughout the simulation. Finally, the monthly Holocene temperature values were interpolated to daily values while total monthly precipitation was distributed randomly among the number (minimum 10) of rainy days per month. For cloudiness, the monthly CRU values from the years 1901-1930 were repeated for the entire simulation period.

The second, historical phase ran from 1901 until 2000. During this period, we forced the model with the CRU TS 3.0 global gridded climate data set (Mitchell and Jones, 2005). Finally, the future scenario phase (see Section 2.3.2) ran from 2001 until 2100, and the climate fields were extracted from RCP8.5 scenario for each location.

2. I have some difficulties with those C accumulation terms as described in lines 217-221 in OM and in Figure 2. I am familiar with but personally don't really like those acronyms. I think some are more useful than other. I think that LARCA is useful as it also represents long-term (10 kyr in this case) actual/"true" average C accumulation rates – which is equivalent to the mean time-weighted C rate from individual cores or synthesis products as cited in the paper. LARCA also should be the same as overall net C balance as discussed in Yu (2011), due to the same peat mass (conservation)

through the last 10 kyr. So LARCA can be used to compare with present and future C accumulation (sequestration) rates. The difference in apparent and actual/"true" C accumulation rates was also discussed in Spahni et al. (2013) and Frolking et al. (2014).

However, I find ARCA is problematic, as it is actually a metric for apparent C accumulation rates – that is, dM/dT (T = 30-year period), despite its name "actual (net) rate of C accumulation". The true C balance rate should consider the decomposition C loss of the entire peat profile during that 30-year period. Am I missing something? A "true" C accumulation rate considering decomposition of previous accumulated peat is needed to compare with past 10 kyr (LARCA) and future C rates. I find that the difference between FLARCA and LARCA is a useful metric to quantify the average true C accumulation rates in the future, so that should be the metric showing in Table 1 and discussed. Both NFRCA and RERCA are apparent C accumulation rates and may not be as useful. I don't think RERCA has been discussed much in the manuscript. I suggest they can be removed from discussion.

Response: We agree that the LARCA is a more useful metric than the other carbon accumulation terms but all of them give different information about the peatland carbon accumulation rates. We have not included RERCA in Table 1 because we think it adds little to the existing information. For ARCA calculation, we do take into account the previous decomposition loss of the entire peat profile. So, it is not an apparent rate, but the actual one. The same is true of NFRCA, which allows a comparison of how different regions might behave in the future (Table 1, Figure 8). It is also a useful metric for readers wishing to compare our findings with those of other studies so we would prefer to keep this in the manuscript. The difference between FLARCA and LARCA is already present in Figure 7 and discussed in the text as well (see lines 485-490 and 545 to 550 in the OM).

3. In general, the manuscript is well written and clearly organized. However, I find some text in subsection 3.2 belong to Discussion, rather than Results section. For example,

lines 315-322 and lines 336-339. Perhaps these can be moved to Discussion section.

Response: We feel that the text in lines 315-322 and lines 336-339 are more suitable for the Results section because we describe results shown in Figure 8. We return to these results in the Discussion section (see lines 564-570 and 394-420 in the OM).

Specific comments:

Title: I suggest to change to ". . .across the pan-Arctic region", as stated in some places of the text. It should change throughout the text.

Response: We have changed "the pan-Arctic" throughout out the text to " the pan-Arctic region".

L22: change to "central and eastern Canada" (lower case)

Response: We agree with this and changed it to a lower case.

L28: change to "higher CO2"?

Response: We agree with this and changed it to higher CO2.

L47: either "organic-rich" or "C-rich", but both together a bit awakward

Response: We have changed it to "C-rich".

Paragraph 1 L57-74: A nice paragraph to summarize peatland models. I'd suggest to concisely mention the following models as well: Spahni et al. (2013), Stocker et al. (2014) and Wang et al. (2016). The first two used an LPJ model version to simulate peatland C dynamics, while Wang et al. (2016) used a different ecosystem model (TEM).

Response: There are many models which included Peatland dynamics in their modelling framework, and we have included a description of those models from which we took the inspiration to develop our model. The functionalities and scope of a representative set of current peatland models (mentioned by the reviewer) are presented in

Table A1 in Paper I (bg-2016-319). We have now summarized these three models in the same paragraph.

L165: add "," after "depth"

Response: We have added a comma "," after depth.

L193: change to "from 45 to 75. . ."

Response: We have changed it to 45 to 75 °N.

L196: defining 0 BP as 1900 is unnecessary and potentially confusing, as conventionally present = 1950 AD. I suggest just to call it 10,000 years before 1900 AD.

Response: We agree with this and we have changed it to 10,000 years before 1900 AD.

L198: describe concisely how Holocene climate input data were generated and prepared. See my general comments above.

Response: We have added a detailed paragraph about the Holocene climate input in the main text. See our response to the general comment 1 above.

L202: are the CO2 concentration simulated or measured from ice cores? If they are ice-core based measurements, then you could just interpolate ice-core data for your purpose, rather than from the data used in UM model, which likely used the ice-core data in the first place. Clarify. Response: We took the same CO2 values used by the UM model in their time slice experiments and linearly interpolated them to yearly values to force our model. We have clarified it in the text below.

Annual CO2 concentration values to force our model from 10 kyr BP to 1850 AD were interpolated from the millennial values used as a boundary condition in the Hadley Centre Unified Model (UM) (Miller et al., 2008) time slice experiments that were run for each millennium from 10 kyr BP to 1850 AD. From the year 1850 to 2000, we used CO2 values from atmospheric or ice core measurements.

L217: see my general comments about these acronyms.

Response: See our response to the general comment 1 above.

L299-300: I don't think the difference between 20.78 and 20.8 should be discussed. Are they robust enough for discussion? It would be useful to present these differences for different zones in Table 1, instead NFRCA. Apparently some zones accumulate more C in the future than other zones, and differences cancel out.

Response: The results show that climate change and $CO_2$ increases can result in many pan-Arctic regions becoming carbon sources while other regions may enhance their sink capacity. Overall, however, the pan-Arctic sink capacity will remain largely unchanged (similar to 2000) by the end of the century (2100), under the high-end scenario (RCP8.5).

We have changed these lines from:

In the FTPC8.5 experiment, where all the drivers were combined„ a marginal decrease in global mean FLARCA (20.78 g C m-2 yr-1) compared with the mean LARCA (20.8 g C m-2 yr-1) (see Fig. 2) was noticed

To

In the FTPC8.5 experiment, where all the drivers were combined, the global mean FLARCA (20.78 g C m-2 yr-1) was largely unchanged from the mean LARCA (20.8 g C m-2 yr-1) (see Fig. 2).

We think that NFRCA is quite informative and it determines how peatlands have been behaving in response to climate change. The difference between FLARCA and LARCA is important and already presented in Figures 7 and 8 (see blue bars).

L306: the value of 53.2+/-37.0 is different from the one (error term) in Table 1. Check this and other values.

Response: Thank you, we have corrected it in the text.

L315-322: move to Discussion?

Response: We have addressed this point above.

L336-339: move to Discussion section?

Response: We have addressed this point above.

L348: ARCA is an apparent C accumulation rate, so comparing it to LARCA is not very meaningful. But if overall decomposition is considered, it would become "true" C rate. See general comment above.

Response: ARCA is not the apparent C accumulation rate, See our response to the general comment 2 above.

L384: change to "litter addition"?

Response: Done.

L546: change to "permafrost, for example in western Siberia. . ."

Response: Done.

L552 and l585: change to "in the future"

Response: Done.

L600: Table 1: -suggest to modify ARCA by considering decomposition loss from all previous peat. That way, it becomes an actual/"true" C accumulation rate. As now, it is a metric for apparent C rate that does not reflect the C sequestration rate and may not be appropriate to compare with LARCA and future C rates. See my general comments.

-Replace NFRCA by (FLARCA minus LARCA)? (FLARCA minus LARCA) represents mean "true" C sequestration rate from 2001 to 2100. See my comments above. -Note for the Zone J, NFRCA is 52.3 +/-19.2, but it was indicated as 52.3+/-37.0 in the text on line 306 (different error terms). Check this and other values.

Response: ARCA takes into account the decomposition loss from all the previous peat layers and it is an actual/true carbon accumulation rate. We clarified this in the text above.

NFRCA is also a good metric to see how peatlands have been behaving in each region. We have already presented FLARCA – LARCA (see Fig. 7 and 8 (see blue bars)).

Thanks, we have corrected the NFRCA values for Zone J.

Figure 2: -suggest to redefine ARCA by considering decomposition C loss of all previous peat, and delete RERCA and NFRCA. See general comments above. -For FLARCA: change to (Mt/(t+f))? (add an extra pairs of brackets)

Response: We have redefined the ARCA and removed the RERCA from the text (but we kept it in Figure 2) but we prefer to keep NFRCA as explained above. We have added an extra pair of brackets in Fig. 2.

Figure 3: change X-axis label to "Age (ka BP)"

Response: There are different ways to abbreviate the term past thousand years. We use kyear (cal. BP) throughout the paper, and prefer to keep the same notation in this figure.

Figure 4: redefine ARCA?

Response: We have redefined ARCA in the text.

Figure 5: I find these results are exciting. -I wonder if it is useful to add a panel to show (and discuss) the observed permafrost distribution (with various categories of continuous, discontinuous, etc.). -if so, three panels should be on one row from left to right, panels a, b, c (the same for Fig. 6) -perhaps Figures 5 and 6 can be merged as one figure with 5 or 6 panels in two rows, as they are relevant results and easier to compare.

Response: We have improved the figure taking these points into account.

Figure 7: Again, these results are exciting. -What does the "simulated mean C accumulation rate" mean? Is it apparent C rate or actual/"true" C rate (net C balance) that considers decomposition of all previous peat? -maybe better to put 3 panels in one row and move them closer.

Response: The simulated mean carbon accumulation rate is the long-term carbon accumulation rate (actual/"true" carbon rate). In this figure, we presented the mean of LARCA values from the year 1990 to 2000. We have improved the figure taking these points into account.

---

## Author Comment (AC2) · 24 May 2017

We appreciate the time and effort spent by the editor and the reviewers in reviewing this manuscript. We have addressed all the issues indicated in the review reports. General comments:

Over large parts of the manuscript the reader is wondering what the authors want to tell him or her. This is especially severe in sections 3 and 4, where results are reported and discussed. The section 3.2, first paragraph is one example: Here the authors discuss results of climate change experiments. Reading the paragraph feels like a near end-less list of carbon accumulation rates defined in slightly different ways and for different regions. It is not possible to list all occurrences of lacking clarity, therefore I suggest the authors carefully look at the manuscript and rewrite unclear sections. In addition the

conclusions section is extremely weak and vague in discussing the conclusions, while half of the section consists of an outlook that is out of place in the conclusions section. Finally, the authors only scratch at the surface of the capabilities of their model. LPJGUESS should be able to determine changes in vegetation composition – however these are not discussed in the manuscript. Similarly, the authors lay claim to unique capabilities of their model (Conclusions section) – however the results of these unique capabilities are not actually discussed

Response: We are thankful to the reviewer for these reflections. We deliberately limit the scope of the paper to the modelled peatland carbon accumulation, and believe that extending the scope to covering other aspects of the modelled dynamics, such as vegetation change, would make the analysis too broad and detract from the main C-cycle related findings, exacerbating rather than improving any issues with clarity. The logic of the paper structure is to first report the simulated mean pan-Arctic and regional carbon accumulation rates (CAR) also referring to modelled permafrost extent as a critical mediating factor. We then go on to attribute regional and overall patterns in carbon accumulation to temperature, precipitation, and CO2 concentrations as drivers, enabling a discussion of which driver(s) might play an important role in the future. Our intention was to highlight the role of different climate forcing on the fate of peatland carbon and CARs in different regions across the pan-Arctic. We argue this is a reasonable scope and logic for one paper.

While we have referred to changes in vegetation composition and productivity at various places in the manuscript (e.g. lines 309-315, 321-322, 328-329, 359-362, 385-392, 400-404, 411-420 etc.) we have not done so in detail as our main focus in this study was on the dynamics of peatland carbon accumulation. However, we accept the reviewer's point that changes in vegetation composition should receive more attention in the paper, especially in regard to how it influenced CARs. To this end, we will add a paragraph to the Discussion addressing this issue. We have renamed the final section "Conclusions and outlook" to more adequately reflect its content.

While we fully understand that every paper must stand on its own, we do note that this is the second of two companion papers in the same journal, the other, already published, - Chaudhary et al. (2017);doi: 10.5194/bg-14-2571-2017, describing the model and its evaluation in greater detail, also covering the coupling of carbon cycle to vegetation dynamics, and the unique capabilities of our model in comparison to other peatland models (Table S1 in the original paper).

In addition there are a number of minor issues:

The climate forcing used to drive the Holocene experiment is unclear, the reader needs to read Chaudhary et al. (2016) in order to understand how it was derived. A two sentences summary how it was derived, including the Miller et al reference, would help.

Response: We have now included a more detailed description of the Holocene climate input data.

Revised Text:

Each simulation was run for 10,100 years, and comprised three distinct climate-forcing periods. The first, Holocene phase, lasted from 10 kyr before present (BP) until 0 BP. During this period, the model was forced with daily climate fields (temperature, precipitation and cloudiness) constructed by interpolating between monthly values from the year 10,000 calendar years before present (cal. BP) until the year 1900. The monthly Holocene climate forcing data was prepared by the delta-change method by applying the relative monthly anomalies of temperature and precipitation the nearest GCM grid-cell to the site location to their average monthly values from the CRU TS 3.0 global gridded climate data set (Mitchell and Jones (2005) from the period 1901 to 1930. We then linearly interpolated the values between the millennium time slices to get values for each year of the simulation. This method conserves the interannual variability for temperature and precipitation throughout the simulation. Finally, the monthly Holocene temperature values were interpolated to daily values while total monthly precipitation

was distributed randomly among the number (minimum 10) of rainy days per month. For cloudiness, the monthly CRU values from the years 1901-1930 were repeated for the entire simulation period.

The second, historical phase ran from 1901 until 2000. During this period, we forced the model with the CRU TS 3.0 global gridded climate data set (Mitchell and Jones, 2005). Finally, the future scenario phase (see Section 2.3.2) ran from 2001 until 2100, and the climate fields were extracted from RCP8.5 scenario for each location.

Fig. A1 is a reproduction from MacDonald et al. 2006. It is therefore not needed, the authors can refer to the original figure.

Response: We have removed this figure from the Appendix and referred to the original paper.

Page 8 / line 289: The authors refer to regions with shallow active layers (ALD < 0.1m) and refer to their Fig. 6a. This is impossible to follow, since the Fig. only show ALD 50-100 cm, 100-200, 200-300 and > 300 – the range referred to in the text is not shown

Response: Thank you for pointing this out. We have corrected this in the text.

Revised text: active layers (ALD < 50-100 cm)

Page 9 / line 325: the authors refer to Fig. 5b. I assume they mean 6b?

Response: Thanks, we have corrected it in the text.

Revised text: Our simulations suggest that the significant temperature increase implied by the RCP8.5 future scenario will lead to disappearance or fragmentation of permafrost from the peat soil, and deeper active layers (Fig. 6b).

Page 9, lines 324-329: The paragraph deals with temperature effects. The authors then refer to their Fig. 8 III and IV – however panels III and IV show the precipitation and CO2 effects. Clearly there is some logical error in this paragraph.

Response: CO2 and precipitation effects in Fig. 8 III and IV are mentioned in the context of their role as drivers of plant (and litter) production, offsetting temperature-induced increase in decomposition. We make this link clearer in the revised text:

Our simulations suggest that the significant temperature increase implied by the RCP8.5 future scenario will lead to disappearance or fragmentation of permafrost from the peat soil, and deeper active layers (Fig. 6b). Additional soil water changes resulting from the effects of higher temperatures on evapotranspiration rates could then either suppress of accelerate the decomposition rate at many peatland locations (Fig. 8II). Effects of precipitation changes and rising CO2 concentrations on plant productivity can offset decomposition changes, in terms of effects on peat accumulation rate. In the Siberian (C, D and E) and Alaskan (F) zones, the projected higher decomposition rates are compensated by higher plant productivity due to increases in soil moisture and CO2 fertilization (Fig. 8III and IV); bars), leading to a net increase in CAR by 2100 in this scenario.

Page 11, line 375: The authors write that Loisel et al. (2014) is limited to north of 69°N. However, it is south of 69°N that is meant.

Response: We have changed to the south of 69°N:

Revised text: Furthermore, the dataset is limited to areas south of 69 °N.

Page 11, lines 381: unclear, when moist conditions played a role

Response: We have revised the sentence.

Revised text: Suitable climate and optimal local hydrological conditions influenced by favourable underlying topographical settings accelerated the CAR which led to the formation of large peatland complexes in the pan-Arctic region (Yu et al., 2009). CAR is the balance between biological inputs (litter accumulation) and outputs (decomposition and leaching) and these two important processes are quite sensitive to climate variability (Clymo, 1991).

Page 12, lines 415-418: Is this trend only reported in the literature, or does it also occur in the model?

Response: It is reported in the literature but we have also found this in some of our model evaluation sites in our companion paper (Chaudhary et al., 2017). We have referred to the companion paper and other studies in the manuscript.

Page 14, line 490: the authors refer to Fig. 8 b, c and d, but they mean II, III, and IV

Response: Thanks, we have corrected it.

Revised text: Hence, this region is projected to act as a C sink in the future (Fig. 8 I). It is notable in our simulations that temperature increases in the T8.5 experiment have a very limited overall effect on decomposition rate in Russia (Zones C, D and E) while precipitation and CO2 fertilization have a positive effect on C build up (Fig. 8 II, III and IV).

Fig. 4: What the authors call a "dotted" line in the Figure legend is usually referred to as a "dashed" line. In addition, the black line discussed in the legend is invisible in the Figure.

Response: We have changed that to a "dashed" line in the caption and the black line discussed in the legend is also the same dashed line. We have clarified this in the text.

Revised text: Fig. 4 Simulated Holocene peat accumulation rates across the 10 zones considered in this study (blue dots) and for the pan-Arctic region as a whole (dashed black line). The x axes show the number of sites partitioned into 10 zones. The black dashed line is the pan-Arctic average with standard deviation (black line outside the y-axes) and the red dashed line is the average among zones with standard deviation in light red patch. (I) simulated long-term (apparent) rate of C accumulation (LARCA); (II) simulated actual rate of C accumulation (ARCA) for the last 30 years. Blue bars show the difference between ARCA and LARCA mean values for the respective zone (II-I)

Fig. 6: Choice of colours is less than perfect. 1) Are no active layer depths of less than

50 cm shown? This is implied by Figures 6a and b. 2) the colour scale chosen in Fig. 6c usually implies a symmetric range from positive (green) to negative (red), with no change indicated by yellow. However in this Figure all values are negative.

Response: Thank you, we have improved the figure taking these points into account.

Fig. 7c: Colour scale not symmetric – zero value unclear (see also my comment to Fig. 6, part 2)

Response: Thank you, we have improved the figure taking these points into account.

Fig. 8: The same dashed / dotted issue as in Fig. 4

Response: We have changed it to a "dashed" line.

Fig. A2: Colour scales are not centered around zero and are different between plots, making comparisons very hard.

Response: Thank you for pointing this out. We have improved the figure.

Fig. A3: On an A4 printout, this Figure is still too small to see any details. In addition, there are no axis subdivisions between -0.5/0/0.5, making it extremely hard to read C3 BGD Interactive comment Printer-friendly version Discussion paper anything from the Figure

Response: Thank you for pointing this out. We have improved the figure.

Reference:

Chaudhary, N., Miller, P. A., and Smith, B.: Modelling Holocene peatland dynamics with an individual-based dynamic vegetation model, Biogeosciences, 14, 2571-2596, doi: 10.5194/bg-14-2571-2017, 2017. Loisel, J., Yu, Z. C., Beilman, D. W., Camill, P., Alm, J., Amesbury, M. J., Anderson, D., Andersson, S., Bochicchio, C., Barber, K., Belyea, L. R., Bunbury, J., Chambers, F. M., Charman, D. J., De Vleeschouwer, F., Fialkiewicz-Koziel, B., Finkelstein, S. A., Galka, M., Garneau, M., Hammarlund, D., Hinchcliffe,

W., Holmquist, J., Hughes, P., Jones, M. C., Klein, E. S., Kokfelt, U., Korhola, A., Kuhry, P., Lamarre, A., Lamentowicz, M., Large, D., Lavoie, M., MacDonald, G., Magnan, G., Makila, M., Mallon, G., Mathijssen, P., Mauquoy, D., McCarroll, J., Moore, T. R., Nichols, J., O'Reilly, B., Oksanen, P., Packalen, M., Peteet, D., Richard, P. J. H., Robinson, S., Ronkainen, T., Rundgren, M., Sannel, A. B. K., Tarnocai, C., Thom, T., Tuittila, E. S., Turetsky, M., Valiranta, M., van der Linden, M., van Geel, B., van Bellen, S., Vitt, D., Zhao, Y., and Zhou, W. J.: A database and synthesis of northern peatland soil properties and Holocene carbon and nitrogen accumulation, Holocene, 24, 1028-1042,doi: 10.1177/0959683614538073, 2014. Mitchell, T. D. and Jones, P. D.: An improved method of constructing a database of monthly climate observations and associated high-resolution grids, Int. J. Climatol., 25, 693-712,doi: 10.1002/joc.1181, 2005.

---

## Author Response (AR2)

**Journal:** Biogeosciences

**Manuscript no.:** bg-2017-34

**Title:** Modelling past, present and future peatland carbon accumulation across the pan-Arctic region

**Author(s):** Nitin Chaudhary et al.

**Date submitted:** 16 Feb. 2017

We appreciate the time and effort spent by the editor and the reviewers in reviewing this manuscript. We have addressed all the issues indicated in the review reports.

**Reviewer 1**

**General comments:** 1. I'd suggest that the authors briefly describe how they generate Holocene climate input data to drive the model. I understand that has been described in detail in the previous model description/calibration paper, but it would be useful to provide a concise description in the paragraph (Lines 191-206 in the original manuscript (OM)) as well (such as model- simulated paleoclimate).

> **Response:** We avoided a detailed description of the Holocene climate input data to make the paper more concise, referring instead to our Paper I (Chaudhary et al. (2017); doi: 10.5194/bg-14-2571-2017) but based on the suggestions of the reviewers, we have now included a more detailed description of the Holocene climate input data in this paper (see lines 295-370; page 6-7 in the revised manuscript (RM)).

> **Revised Text:**

> While peatland initiation started at ca. 12-13 kyr BP in high latitude areas, the majority of peatlands formed after 10 kyr BP (MacDonald et al., 2006). Therefore, each simulation was run for 10,100 years, and comprised three distinct climate-forcing periods. The first, Holocene, phase lasted from 10 kyr before present (BP) until 0 BP. During this period, the model was forced with daily climate fields (temperature, precipitation and cloudiness) constructed by interpolating between monthly values from the year 10,000 calendar years before present (cal. BP) until 1900. The monthly Holocene climate forcing data were prepared by the delta-change method by applying relative monthly anomalies of temperature and precipitation for the nearest GCM gridcell (see section 2.3.2) to the site location to their average monthly values from the CRU TS 3.0 global gridded climate data set (Mitchell and Jones, 2005) from the period 1901 to 1930. We then linearly interpolated the values between the millennium time slices to get values for each year of the simulation. This method conserves interannual variability for temperature and precipitation from the baseline historical climate (1901-1930) throughout the simulation. Finally, the monthly Holocene temperature values were interpolated to daily values while total monthly precipitation was distributed randomly among the number (minimum 10) of rainy days per month. For cloudiness, the monthly CRU values from the years 1901-1930

were repeated for the entire simulation period. The second, historical phase ran from 1901 until 2000. During this period, we forced the model with the CRU data. Finally, the future scenario phase (see Section 2.3.2) ran from 2001 until 2100, applying anomalies extracted for the RCP8.5-forced GCM climate fields (section 2.3.2) for each location. Annual $CO_2$ concentration values to force our model from 10 kyr BP to 1850 AD were interpolated from the millennial values used as a boundary condition in the Hadley Centre Unified Model (UM) (Miller et al., 2008) time slice experiments that were run for each millennium from 10 kyr BP to 1850 AD. From the year 1850 to 2000, we used $CO_2$ values from atmospheric or ice core measurements.

2. I have some difficulties with those C accumulation terms as described in lines 217- 221 in the OM and in Figure 2. I am familiar with but personally don't really like those acronyms. I think some are more useful than other. I think that LARCA is useful as it also represents long-term (10 kyr in this case) actual/"true" average C accumulation rates – which is equivalent to the mean time-weighted C rate from individual cores or synthesis products as cited in the paper. LARCA also should be the same as overall net C balance as discussed in Yu (2011), due to the same peat mass (conservation) through the last 10 kyr. So LARCA can be used to compare with present and future C accumulation (sequestration) rates. The difference in apparent and actual/"true" C accumulation rates was also discussed in Spahni et al. (2013) and Frolking et al. (2014).

However, I find ARCA is problematic, as it is actually a metric for apparent C accumulation rates – that is, dM/dT (T = 30-year period), despite its name "actual (net) rate of C accumulation". The true C balance rate should consider the decomposition C loss of the entire peat profile during that 30-year period. Am I missing something? A "true" C accumulation rate considering decomposition of previous accumulated peat is needed to compare with past 10 kyr (LARCA) and future C rates.

I find that the difference between FLARCA and LARCA is a useful metric to quantify the average true C accumulation rates in the future, so that should be the metric showing in Table 1 and discussed. Both NFRCA and RERCA are apparent C accumulation rates and may not be as useful. I don't think RERCA has been discussed much in the manuscript. I suggest they can be removed from discussion.

**Response:**

We agree that the LARCA is a more useful metric than the other carbon accumulation terms but all of them give different information about the peatland carbon accumulation rates. We have not included RERCA in Table 1 (on page 19) because we think it adds little to the existing information. For ARCA calculation, we do take into account the previous decomposition loss of the entire peat profile. So, it is not an apparent rate, but the actual one. The same is true of NFRCA, which allows a comparison of how different regions might behave in the future (Table 1; page 19). It is also a useful metric for readers wishing to compare our findings with those of other studies so we would prefer to keep this in the manuscript. The difference between FLARCA and LARCA is already present in Figure 8 (now Fig. 7; blue bars in the RM) and discussed in the text as well (see lines 511-515 in the RM).

3. In general, the manuscript is well written and clearly organized. However, I find some text in subsection 3.2 belong to Discussion, rather than Results section. For example, lines 315-322 and lines 336-339. Perhaps these can be moved to Discussion section.

>   **Response:** We feel that the text in lines 315-322 (now lines 592-600; page 10 in the RM) and lines 336-339 (now lines 615-620; page 10 in the RM) are more suitable for the Results section because we describe results shown in Figure 8 (now Fig. 7 in the RM). We return to these results in the Discussion section (see lines 1016-1034; page 16-17 and lines: 763-787; page 12 in the RM).

**Specific comments:**

Title: I suggest to change to ". . .across the pan-Arctic region", as stated in some places of the text. It should change throughout the text.

>   **Response:** We have changed "the pan-Arctic" throughout out the text to " the pan-Arctic region" (see line 2, 18, 99, 103, 293 and 450 in the RM)

L22: change to "central and eastern Canada" (lower case)

>   **Response:** We agree with this and changed it to a lower case (see line 20; page 1 in the RM).

L28: change to "higher CO2"?

>   **Response:** We agree with this and changed it to higher $CO_2$ (see line 26; page 1 in the RM).

  L47: either "organic-rich" or "C-rich", but both together a bit awakward

>   **Response:** We have changed it to "C-rich" (see line 53; page 2 in the RM).

Paragraph l 57-74: A nice paragraph to summarize peatland models. I'd suggest to concisely mention the following models as well: Spahni et al. (2013), Stocker et al. (2014) and Wang et al. (2016). The first two used an LPJ model version to simulate peatland C dynamics, while Wang et al. (2016) used a different ecosystem model (TEM).

>   **Response:** There are many models which included peatland dynamics in their modelling framework, and we have included description of those models from which we took the inspiration to develop our model. The functionalities and scope of a representative set of current peatland models (mentioned by the reviewer) are presented in Table S1 in Paper I (bg-2016-319). We have summarized these three and some other models in the same paragraph (see lines 81-90; page 2-3 in the RM).

**Revised text:** Other model representations have also included peatland processes in their frameworks (Morris et al., 2012; Alexandrov et al., 2016; Wu et al., 2016) and been shown to perform reasonably at different sites. In addition, some of these models have been applied over large areas (Kleinen et al., 2012; Schuldt et al., 2013; Stocker et al., 2014; Alexandrov et al., 2016) to simulate regional peatland dynamics. Lately, Chaudhary et al. 2017 has included

a new implementation of peatland and permafrost dynamics with the representation of spatial heterogeneity in the dynamic vegetation model (LPJ-GUESS) and shown to capture reasonable peat accumulation, permafrost dynamics and vegetation distribution at Stordalen site in the north of Sweden.

L165: add "," after "depth"

  **Response:** We have added a comma "," after depth (see line 230; page 5 in the RM).

L193: change to "from 45 to 75. . ."

  **Response:** We have changed it to 45 to 75 °N (see line 294; page 6 in the RM).

L196: defining 0 BP as 1900 is unnecessary and potentially confusing, as conventionally present = 1950 AD. I suggest just to call it 10,000 years before 1900 AD.

  **Response:** We agree with this and we have changed it to 10,000 years before 1900 AD (see lines: 300-301; page 6 in the RM).

L198: describe concisely how Holocene climate input data were generated and prepared. See my general comments above.

  **Response:** We have added a detailed paragraph about the Holocene climate input in the main text (see lines 295-370; page 6-7 in the RM). See our response to the general comment 1 above.

L202: are the $CO_2$ concentration simulated or measured from ice cores? If they are ice-core based measurements, then you could just interpolate ice-core data for your purpose, rather than from the data used in UM model, which likely used the ice-core data in the first place. Clarify.

  **Response:** We took the same $CO_2$ values used by the UM model in their time slice experiments and linearly interpolated them to yearly values to force our model. We have clarified it in the text below (see lines 313-370; page 6-7 in the RM).

  **Revised text:** Annual $CO_2$ concentration values to force our model from 10 kyr BP to 1850 AD were interpolated from the millennial values used as a boundary condition in the Hadley Centre Unified Model (UM) (Miller et al., 2008) time slice experiments that were run for each millennium from 10 kyr BP to 1850 AD. From the year 1850 to 2000, we used $CO_2$ values from atmospheric or ice core measurements.

L217: see my general comments about these acronyms.

  **Response:** See our response to the general comment 1 above.

L299-300: I don't think the difference between 20.78 and 20.8 should be discussed. Are they robust enough for discussion? It would be useful to present these differences for different zones in Table 1, instead NFRCA. Apparently some zones accumulate more C in the future

than other zones, and differences cancel out.

**Response:** The results show that climate change and $CO_2$ increases can result in many pan-Arctic regions becoming carbon sources while other regions may enhance their sink capacity. Overall, however, the pan-Arctic sink capacity will remain largely unchanged (similar to 2000) by the end of the century (2100), under the high-end scenario (RCP8.5).

We have changed these lines from:

In the FTPC8.5 experiment, where all the drivers were combined, a marginal decrease in global mean FLARCA (20.78 g C $m^{-2}$ $yr^{-1}$) compared with the mean LARCA (20.8 g C $m^{-2}$ $yr^{-1}$) (see Fig. 2) was noticed

To

In the FTPC8.5 experiment, where all the drivers were combined, the global mean FLARCA (20.78 g C $m^{-2}$ $yr^{-1}$) was largely unchanged from the mean LARCA (20.8 g C $m^{-2}$ $yr^{-1}$) (see Fig. 2)- (see lines 511-514; page 9 in the RM).

We think that NFRCA is quite informative and it determines how peatlands have been behaving in response to climate change. The difference between FLARCA and LARCA is important and already presented in Figures 7 and 8 (now Figs. 6 and 7; page-29-30 and also see blue bars in Fig. 7).

L306: the value of 53.2+/-37.0 is different from the one (error term) in Table 1. Check this and other values.

**Response:** Thank you, we have corrected it in the text (line 518; page 9 in the RM).

L315-322: move to Discussion?

**Response:** We have addressed this point above. See our response to the general comment 3.

L336-339: move to Discussion section?

**Response:** We have addressed this point above. See our response to the general comment 3.

L348: ARCA is an apparent C accumulation rate, so comparing it to LARCA is not very meaningful. But if overall decomposition is considered, it would become "true" C rate. See general comment above.

**Response:** ARCA is not the apparent C accumulation rate. See our response to the general comment 2 above.

L384: change to "litter addition"?

**Response:** Done (line 381-382; page 7 in the RM).

L546: change to "permafrost, for example in western Siberia. . ."

**Response:** Done (line 998; page 16 in the RM).

L552 and l585: change to "in the future"

**Response:** Done (line 1004; page 16 and line 1048; page17 in the RM).

L600: Table 1: -suggest to modify ARCA by considering decomposition loss from all previous peat. That way, it becomes an actual/"true" C accumulation rate. As now, it is a metric for apparent C rate that does not reflect the C sequestration rate and may not be appropriate to compare with LARCA and future C rates. See my general comments.

-Replace NFRCA by (FLARCA minus LARCA)? (FLARCA minus LARCA) represents mean "true" C sequestration rate from 2001 to 2100. See my comments above.

-Note for the Zone J, NFRCA is 52.3 +/-19.2, but it was indicated as 52.3+/-37.0 in the text on line 306 (different error terms). Check this and other values.

**Response:** ARCA takes into account the decomposition loss from all the previous peat layers and it is an actual/true carbon accumulation rate. We clarified this in the text above.

NFRCA is also a good metric to see how peatlands have been behaving in each region. We have already presented FLARCA – LARCA (see Fig. 7 (see blue bars); page 30).

Thanks, we have corrected the NFRCA values for Zone J in the text (line 518; page 9 in the RM).

Figure 2: -suggest to redefine ARCA by considering decomposition C loss of all previous peat, and delete RERCA and NFRCA. See general comments above. -For FLARCA: change to (Mt/(t+f))? (add an extra pairs of brackets)

**Response:** ARCA is not the apparent C accumulation rate, but the actual one. See our response to the general comment 2 above. We clarified it in the text. We have removed the RERCA from the text (but we kept it in Figure 2) but we prefer to keep NFRCA as explained above. We have added an extra pair of brackets in Fig. 2 on page 25.

Figure 3: change X-axis label to "Age (ka BP)"

**Response:** There are different ways to abbreviate the term past thousand years. We use kyear (cal. BP) throughout the paper, and prefer to keep the same notation in this figure.

Figure 4: redefine ARCA?

**Response:** We have redefined ARCA in the text (see Fig. 4; page 27 in the RM).

Figure 5: I find these results are exciting. -I wonder if it is useful to add a panel to show (and discuss) the observed permafrost distribution (with various categories of continuous, discontinuous, etc.). -if so, three panels should be on one row from left to right, panels a, b, c (the same for Fig. 6) -perhaps Figures 5 and 6 can be merged as one figure with 5 or 6 panels in two rows, as they are relevant results and easier to compare.

    **Response:** We have improved the figure taking these points into account (see Fig. 5; page 28 in the RM).

Figure 7: Again, these results are exciting. -What does the "simulated mean C accumulation rate" mean? Is it apparent C rate or actual/"true" C rate (net C balance) that considers decomposition of all previous peat? -maybe better to put 3 panels in one row and move them closer.

    **Response:** In this figure, we presented the actual mean carbon accumulation values from (a) the year 1990 to 2000 and (b) 2090 to 2100 and the panel (c) shows a difference between (b) and (a). We have improved the figure taking these points into account (see Fig. 6; page 29 in the RM).

**Reviewer 2**

**General comments:**

Over large parts of the manuscript the reader is wondering what the authors want to tell him or her. This is especially severe in sections 3 and 4, where results are reported and discussed. The section 3.2, first paragraph is one example: Here the authors discuss results of climate change experiments. Reading the paragraph feels like a near endless list of carbon accumulation rates defined in slightly different ways and for different regions. It is not possible to list all occurrences of lacking clarity, therefore I suggest the authors carefully look at the manuscript and rewrite unclear sections. In addition the conclusions section is extremely weak and vague in discussing the conclusions, while half of the section consists of an outlook that is out of place in the conclusions section. Finally, the authors only scratch at the surface of the capabilities of their model. LPJGUESS should be able to determine changes in vegetation composition – however these are not discussed in the manuscript. Similarly, the authors lay claim to unique capabilities of their model (Conclusions section) – however the results of these unique capabilities are not actually discussed

> **Response:** We are thankful to the reviewer for these reflections. We deliberately limit the scope of the paper to the modelled peatland carbon accumulation, and believe that extending the scope to covering other aspects of the modelled dynamics, such as vegetation change, would make the analysis too broad and detract from the main C-cycle related findings, exacerbating rather than improving any issues with clarity. The logic of the paper structure is to first report the simulated mean pan-Arctic and regional carbon accumulation rates (CAR) also referring to modelled permafrost extent as a critical mediating factor. We then go on to attribute regional and overall patterns in carbon accumulation to temperature, precipitation, and $CO_2$ concentrations as drivers, enabling a discussion of which driver(s) might play an important role in the future. Our intention was to highlight the role of different climate forcing on the fate of peatland carbon and CARs in different regions across the pan-Arctic. We argue this is a reasonable scope and logic for one paper. We have referred to changes in vegetation composition and productivity at various places in the original manuscript (e.g. lines 309-315, 321-322, 328-329, 359-362, 385-392, 400-404, 411-420 etc.). We have renamed the final section "Conclusions and outlook" to more adequately reflect its content.
>
> While we fully understand that every paper must stand on its own, we do note that this is the second of two companion papers in the same journal, the other, already published, - Chaudhary et al. (2017);doi: 10.5194/bg-14-2571-2017, describing the model and its evaluation in greater detail, also covering the coupling of carbon cycle to vegetation dynamics, and the unique capabilities of our model in comparison to other peatland models (Table S1 in that paper).

**In addition there are a number of minor issues:**

The climate forcing used to drive the Holocene experiment is unclear, the reader needs to read Chaudhary et al. (2016) in order to understand how it was derived. A two sentences summary how it was derived, including the Miller et al reference, would help.

**Response:** We have now included a more detailed description of the Holocene climate input data (see lines 295-370; page 6-7 in the revised manuscript (RM))

**Revised Text:**

While peatland initiation started at ca. 12-13 kyr BP in high latitude areas, the majority of peatlands formed after 10 kyr BP (MacDonald et al., 2006). Therefore, each simulation was run for 10,100 years, and comprised three distinct climate-forcing periods. The first, Holocene, phase lasted from 10 kyr before present (BP) until 0 BP. During this period, the model was forced with daily climate fields (temperature, precipitation and cloudiness) constructed by interpolating between monthly values from the year 10,000 calendar years before present (cal. BP) until 1900. The monthly Holocene climate forcing data were prepared by the delta-change method by applying relative monthly anomalies of temperature and precipitation for the nearest GCM gridcell (see section 2.3.2) to the site location to their average monthly values from the CRU TS 3.0 global gridded climate data set (Mitchell and Jones, 2005) from the period 1901 to 1930. We then linearly interpolated the values between the millennium time slices to get values for each year of the simulation. This method conserves interannual variability for temperature and precipitation from the baseline historical climate (1901-1930) throughout the simulation. Finally, the monthly Holocene temperature values were interpolated to daily values while total monthly precipitation was distributed randomly among the number (minimum 10) of rainy days per month. For cloudiness, the monthly CRU values from the years 1901-1930 were repeated for the entire simulation period. The second, historical phase ran from 1901 until 2000. During this period, we forced the model with the CRU data. Finally, the future scenario phase (see Section 2.3.2) ran from 2001 until 2100, applying anomalies extracted for the RCP8.5-forced GCM climate fields (section 2.3.2) for each location. Annual $CO_2$ concentration values to force our model from 10 kyr BP to 1850 AD were interpolated from the millennial values used as a boundary condition in the Hadley Centre Unified Model (UM) (Miller et al., 2008) time slice experiments that were run for each millennium from 10 kyr BP to 1850 AD. From the year 1850 to 2000, we used $CO_2$ values from atmospheric or ice core measurements.

Fig. A1 is a reproduction from MacDonald et al. 2006. It is therefore not needed, the authors can refer to the original figure.

**Response:** We have removed this figure from the Appendix and referred the original paper (see lines 295-297; page 6 in the RM)

Page 8 / line 289 (in the original manuscript): The authors refer to regions with shallow active layers (ALD < 0.1m) and refer to their Fig. 6a. This is impossible to follow, since the Fig. only show ALD 50-100 cm, 100-200, 200-300 and > 300 – the range referred to in the text is not shown

**Response:** Thank you for pointing this out. We have corrected this in the text (see line 503; page 9 in the RM).

**Revised text**: active layers (ALD < 100 cm)

Page 9 / line 325: the authors refer to Fig. 5b. I assume they mean 6b?

**Response:** Thanks, we have corrected it in the text and we have combined the Figs. 5 and 6 (see line 603; page 10 and Fig. 5 in the RM).

**Revised text:** Our simulations suggest that the significant temperature increase implied by the RCP8.5 future scenario will lead to disappearance or fragmentation of permafrost from the peat soil, and deeper active layers (Fig. 5b and e).

Page 9, lines 324-329: The paragraph deals with temperature effects. The authors then refer to their Fig. 8 III and IV – however panels III and IV show the precipitation and CO2 effects. Clearly there is some logical error in this paragraph.

**Response:** $CO_2$ and precipitation effects in Fig. 8 III and IV (now Fig. 7) are mentioned in the context of their role as drivers of plant (and litter) production, offsetting temperature-induced increase in decomposition (see lines 601-609; page 10 in the RM). We make this link clearer in the revised text:

**Revised text:** Our simulations suggest that the significant temperature increase implied by the RCP8.5 future scenario will lead to disappearance or fragmentation of permafrost from the peat soil, and deeper active layers (Fig. 5b and e). Additional soil water changes resulting from the effects of higher temperatures on evapotranspiration rates could then either suppress or accelerate the decomposition rate at many peatland locations (Fig. 7 II). Effects of precipitation changes and rising $CO_2$ concentrations on plant productivity can offset decomposition changes, in terms of effects on peat accumulation rate. In the Siberian (C, D and E) and Alaskan (F) zones, the projected higher decomposition rates are compensated by higher plant productivity due to increases in soil moisture and $CO_2$ fertilization (Fig. 7 III and IV); bars), leading to a net increase in CAR by 2100 in this scenario.

Page 11, line 375: The authors write that Loisel et al. (2014) is limited to north of 69°N. However, it is south of 69°N that is meant.

**Response:** We have changed to the south of 69°N (see line 727; page 11 in the RM):

**Revised text:** Furthermore, the dataset is limited to areas south of 69 °N.

Page 11, lines 381: unclear, when moist conditions played a role

**Response:** We have revised the sentence (see lines 752; page 11 in the RM).

**Revised text:** Suitable climate and optimal local hydrological conditions influenced by favourable underlying topographical settings accelerated the CAR which led to the formation of large peatland complexes in the pan-Arctic region (Yu et al., 2009). CAR is the balance between biological inputs (litter addition) and outputs (decomposition and leaching) and these two important processes are quite sensitive to climate variability (Clymo, 1991).

Page 12, lines 415-418: Is this trend only reported in the literature, or does it also occur in the model?

**Response:** It is reported in the literature but we have also found this in some of our model evaluation sites in our companion paper (Chaudhary et al., 2017). We have referred to the companion paper and other studies in the manuscript (see lines 774-779; page 12 in the RM).

**Revised text:** In our scenario simulations (Table 2), we find that higher temperature leads to thawing of permafrost that in turn increases the moisture availability, at least initially. The rise in temperature also results in early spring snowmelt and a longer growing season (Euskirchen et al., 2006) while, in the same time frame, atmospheric $CO_2$ concentration will also increase. These factors lead to increases in plant productivity, leading to higher CAR (Klein et al., 2013; Chaudhary et al., 2017), even in cases where moisture- and temperature-driven peat decomposition also speeds up.

Page 14, line 490: the authors refer to Fig. 8 b, c and d, but they mean II, III, and IV

**Response:** Thanks, we have corrected it (see lines 920-923; page 14 in the RM).

**Revised text:** Hence, this region is projected to act as a C sink in the future (Fig. 7 I). It is notable in our simulations that temperature increases in the T8.5 experiment have a very limited overall effect on decomposition rate in Russia (zones C, D and E) while precipitation and $CO_2$ fertilization have a positive effect on C build up (Fig. 7 II, III and IV).

Fig. 4: What the authors call a "dotted" line in the Figure legend is usually referred to as a "dashed" line. In addition, the black line discussed in the legend is invisible in the Figure.

**Response:** We have changed that to a "dashed" line in the caption and the black line discussed in the legend is also the same dashed line. We have clarified this in the text (see Fig. 4; page 27 in the RM).

**Revised text: Fig. 4** Simulated Holocene peat accumulation rates across the 10 zones considered in this study (blue dots) and for the pan-Arctic region as a whole (dashed black line). The x axes show the number of sites partitioned into 10 zones. The black dashed line is the pan-Arctic average with standard deviation (black line outside the y-axes) and the red dashed line is the average among zones with standard deviation in light red patch. (I) simulated long-term (apparent) rate of C accumulation (LARCA); (II) simulated actual (true) rate of C accumulation (ARCA) for the last 30 years. Blue bars

show the difference between ARCA and LARCA mean values for the respective zone (II-I).

Fig. 6: Choice of colours is less than perfect. 1) Are no active layer depths of less than 50 cm shown? This is implied by Figures 6a and b. 2) the colour scale chosen in Fig. 6c usually implies a symmetric range from positive (green) to negative (red), with no change indicated by yellow. However in this Figure all values are negative.

Fig. 7c: Colour scale not symmetric – zero value unclear (see also my comment to Fig. 6, part 2)

**Response:** Thank you, we have improved the figures and taking these points into account. We have combined the figure 5 and 6 (see page 28). We have included more categories in Figures 6 (a) and (b) (now 5d and e). We have now used a new colour scale with no change indicated by dark green. We have improved the symmetry in the colour scale in the Fig 7(c) (now 6(c)) (see page 29).

Fig. 8: The same dashed / dotted issue as in Fig. 4

**Response:** We have changed it to a "dashed" line (see Fig. 7 on page 30 in the RM).

Fig. A2: Colour scales are not centered around zero and are different between plots, making comparisons very hard.

**Response:** Thank you for pointing this out. We have improved the figure (see Fig. A1; page 31 in the RM).

Fig. A3: On an A4 printout, this Figure is still too small to see any details. In addition, there are no axis subdivisions between -0.5/0/0.5, making it extremely hard to read C3 BGD Interactive comment Printer-friendly version Discussion paper anything from the Figure

**Response:** We have improved the figure (see Fig. A2 on page 32 in the RM).